# Performance-optimized deep neural networks are evolving into worse models of inferotemporal visual cortex

**Drew Linsley**[*1,2], **Iván F. Rodriguez**[*1], **Thomas Fel**[3], **Michael Arcaro**[4],
**Saloni Sharma**[5], **Margaret Livingstone**[5], **Thomas Serre**[1,2,3]
`drew_linsley,ivan_felipe_rodriguez@brown.edu`

## Abstract

One of the most impactful findings in computational neuroscience over the past decade is that the object recognition accuracy of deep neural networks (DNNs) correlates with their ability to predict neural responses to natural images in the inferotemporal (IT) cortex [1, 2]. This discovery supported the long-held theory that object recognition is a core objective of the visual cortex, and suggested that more accurate DNNs would serve as better models of IT neuron responses to images [3–5]. Since then, deep learning has undergone a revolution of scale: billion parameter-scale DNNs trained on billions of images are rivaling or outperforming humans at visual tasks including object recognition. Have today's DNNs become more accurate at predicting IT neuron responses to images as they have grown more accurate at object recognition?

Across three independent experiments, we find this is not the case: DNNs have become progressively worse models of IT as their accuracy has increased on ImageNet. To understand why DNNs experience this trade-off and evaluate if they are still an appropriate paradigm for modeling the visual system, we turn to recordings of IT that capture spatially resolved maps of neuronal activity elicited by natural images [6]. These neuronal activity maps reveal that DNNs trained on ImageNet learn to rely on different visual features than those encoded by IT and that this problem worsens as their accuracy increases. We successfully resolved this issue with the *neural harmonizer*, a plug-and-play training routine for DNNs that aligns their learned representations with humans [7]. Our results suggest that harmonized DNNs break the trade-off between ImageNet accuracy and neural prediction accuracy that assails current DNNs and offer a path to more accurate models of biological vision. Our work indicates that the standard approach for modeling IT with task-optimized DNNs needs revision, and other biological constraints, including human psychophysics data, are needed to accurately reverse-engineer the visual cortex.

## 1 Introduction

The release of AlexNet [8] was significant not only for shifting the paradigm of computer vision into the era of deep learning, it also heralded a new approach for "systems identification" and

---

[*]These authors contributed equally.

[1]Department of Cognitive, Linguistic, & Psychological Sciences, Brown University, Providence, RI

[2]Carney Institute for Brain Science, Brown University, Providence, RI

[3]Artificial and Natural Intelligence Toulouse Institute, Toulouse, France

[4]Department of Psychology, University of Pennsylvania, Philadelphia, PA

[5]Harvard Medical School, Cambridge, MA

37th Conference on Neural Information Processing Systems (NeurIPS 2023).

approximating the transformations used by neurons in the visual cortex to build robust and invariant object representations. Over the past decade, deep neural networks (DNNs) like AlexNet, which are trained for object recognition on ImageNet [9], have been found to contain units that significantly better fit neural activity in inferior temporal (IT) cortex in non-human primates compared to classic hand-tuned models from computational neuroscience [1]. It was later found that DNN predictions of neural data improved as these models grew more accurate at object recognition and began to rival humans on the task [10, 11]. This surprising similarity between such task-optimized DNNs and brains supported the extant theory that object recognition is a core principle shaping the organization of the visual cortex [12], and raised the question of how important the many biological details amassed by visual neuroscience actually are for predicting neural responses in visual cortex. In the years since those findings, deep learning has undergone a revolution of scale, and current DNNs which rival or exceed human accuracy in vision and language are significantly larger and trained with orders of magnitude more data than ever before [13]. Does the object recognition accuracy of a DNN still correlate with its ability to predict IT responses to natural objects?

To answer this question, we turned to Brain-Score [3], the standard approach for benchmarking the accuracy of models at predicting neural activity in visual cortex of non-human primates. In brief, the Brain-Score evaluation method involves linearly mapping model unit responses to neural activity and then evaluating model unit predictions on held-out images. With this approach, we found a consistent trend across three different IT datasets hosted on the official Brain-Score website (`Brain-score.org`), which reflect neural responses to gray-scale versions of realistic rendered objects and natural images [14–16]. As DNNs have improved on ImageNet [9] over recent years, they have become progressively less accurate at predicting IT neuron responses to images (Fig. 1) *

In this study, we investigated two potential explanations for why task-optimized DNNs are turning into poor models of IT. (*i*) The internet data diets of DNNs and the routines used to train them to high accuracy on ImageNet leads them to learn the wrong visual features for explaining primate vision and IT responses to objects [7, 17–19]. (*ii*) Newer DNNs are becoming less brain-like in their architectures, and this problem has been magnified as DNNs have grown larger and ultimately deeper than the visual cortex [20].

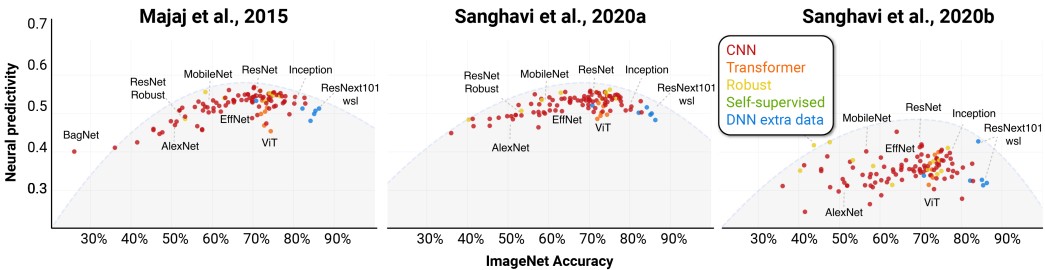

Figure 1: **Deep neural networks (DNNs) are becoming progressively worse models of inferior temporal (IT) cortex as they grow more accurate on ImageNet.** Experimental data shown here is taken from `Brain-score.org`, and each experiment utilized different stimuli. The 104 dots in each panel depict the ImageNet accuracy and neural prediction accuracy of DNNs, and the grey-shaded region denotes the pareto-front governing the trade-off between these variables. EffNet=EfficientNet.

**Contributions.** To understand if the trade-off between ImageNet accuracy and IT predictions that we observed (Fig. 1) is due to the training routines and data diets of DNNs or their architectures, we turned to a new set of experimental recordings of IT neuronal responses to high-resolution color images [6]. The experimental images were significantly closer to the statistical distribution of images in ImageNet (Fig. S1.), unlike prior studies of IT [14]. The recordings also provided a coarse estimate of which image features drove neuronal activity (Fig. 2), which helped characterize DNN errors in explaining neural responses and explain why DNNs are becoming worse models of IT. We adopted the Brain-Score evaluation method to measure the prediction accuracy of 135 DNNs on recordings from medial (ML) and posterior (PL) lateral IT in two different Monkeys. To summarize our findings:

---

*In [11] it was noted that while there was an overall correlation of 0.92 between DNN accuracy on ImageNet and predicting IT responses, the correlation weakened for state-of-the-art models.

- We observed the same trade-off we found on `Brain-Score.org` data (Fig. 1) in each of our recordings: DNNs are becoming less accurate at predicting IT responses as they improve on ImageNet (Fig. 3).

- DNNs trained on ImageNet learn different features than those encoded by IT (Fig. 4), and this mismatch is not helped by training on more internet data, using newer DNN architectures like Transformers, relying on self-supervision, or optimizing for adversarial robustness.

- We successfully broke this trade-off by training DNNs with the *neural harmonizer* [7] and aligning the representations they learn for object recognition with humans.

- We further demonstrate that harmonized DNNs (hDNNs) are not only significantly better at explaining IT responses than any other DNN available, they also generate interpretable hypotheses on the features driving IT neuron responses.

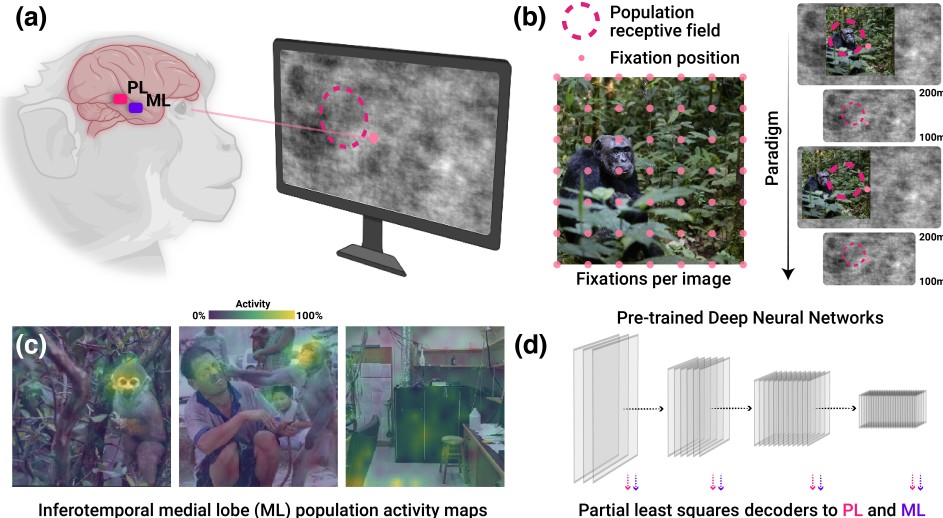

Figure 2: **IT recordings that reveal spatial maps of neuronal responses to complex natural images offer unprecedented insights into their feature selectivity [6]. (a)** Neurons in posterior (PL) and/or medial (ML) lateral IT in two animals were localized using functional magnetic resonance imaging (fMRI), and neural responses to images were recorded using chronically implanted 32-channel multi-electrode arrays. **(b)** The monkeys were rapidly shown each image multiple times for 200ms each time (Monkey 1: $n = 256$, Monkey 2: $n = 49$). Images were positioned differently each time to measure neural responses to every part of the image. **(c)** This procedure yielded spatially resolved maps of neural activity for an image, revealing the relative importance of different features for the recorded neurons. **(d)** DNNs were fit to these recordings following the Brain-Score evaluation method [11], in which partial least squares decoders were used to find the units in a DNN that provided the best match for neuronal responses to images.

## 2 Methods

**Neural recordings.** We leveraged recordings of IT neuronal responses from two monkeys [6], which were designed to reveal feature preferences of putative face-selective neurons. The recorded neurons also exhibited selectivity for non-face object features [6]. Recordings were made using chronically implanted 32-channel multi-electrode arrays within the fMRI-defined middle lateral (ML) and posterior lateral (PL) face patches of one monkey (Monkey 1), and ML of another monkey (Monkey 2, Fig. 2a). The activating regions in these areas were mapped (1-3$^o$), then each image was shown to the animals after they were cued to fixate at a specific position in space (Fig. 2b). The same images were shown multiple times while the monkeys fixated at a red dot in the center of the screen, which made it possible to derive spatial activity maps of neuronal responses (Fig. 2c). Fixation positions fell in a $16 \times 16$ grid for Monkey 1 and $7 \times 7$ grid for Monkey 2, and the average neuronal activity at each position was taken to generate spatial activity maps. A total of 14 images were shown to Monkey 1, and a different set of 14 images were shown to Monkey 2.

The recordings for Monkey 1 resulted in responses from 32 neurons in ML and 31 neurons in PL. For Monkey 2, we obtained responses from 32 neurons (see Appendix B for more details). Following [6], we binned neuronal responses every 40ms of the recording, from 50ms to 250ms. Within each bin, we calculated the noise ceiling for every neuron, which represents the maximum correlation achievable between any two neurons within that time interval. Noise ceilings for each recording were similar to those reported for standard IT Brain-Score datasets (Appendix B). In the main text, we present modeling results from the time bins that exhibited the highest average noise ceiling for each Monkey and recording site. Additional results from other time bins, which are consistent with these findings, can be found in Appendix E.

**DNNs.** We investigated the neural fits of 135 different DNNs representing a variety of approaches used in computer vision today: 62 convolutional neural networks trained on ImageNet [21–34, 34–47] (CNNs), 23 DNNs trained on other datasets in addition to ImageNet (which we refer to as "DNN extra data") [32, 36, 48], 25 vision transformers [49–54] (ViTs), 10 DNNs trained with self-supervision [55, 56], and 15 DNNs trained to be robust to noise or adversarial examples [57, 58]. Each model was implemented in PyTorch with the TIMM toolbox (`https://github.com/huggingface/pytorch-image-models`) and pre-trained weights. Inference was executed on one NVIDIA TITAN X GPU. Additional model details, including the licenses used for each, are detailed in Appendix C

**Neural Harmonizer.** It was recently found that DNN representations are becoming less aligned with human perception as they evolve and improve on ImageNet [7, 59, 60]. A partial solution to this problem is the *neural harmonizer*, a training routine that can be combined with any DNN to align its representations with humans without sacrificing performance. Here, we test the hypothesis that aligning DNNs with human visual representations can similarly significantly improve their accuracy in predicting neural responses to images.

Training DNNs with the *neural harmonizer* on ImageNet involves an additional loss to standard cross-entropy for object recognition. This extra loss forces a model's gradients to appear as similar as possible to feature importance maps derived from human behavioral experiments (see [7] for details). To implement this loss, let $\mathcal{P}_i(.)$ be a function that a multi-scale Gaussian pyramid of a human feature importance map $\phi$ to $N$, with $i \in \{1, ..., N\}$. During training, we seek to minimize $\sum_i^N ||\mathcal{P}_i(g(f_\theta, x)) - \mathcal{P}_i(\phi)||^2$ in order to align feature importance maps between humans and DNNs at every scale of the pyramid. Before they are compared, feature importance maps from humans and DNNs are normalized and rectified using $z(.)$, a function that transforms a feature importance map $\phi$ from either source to have 0 mean and unit standard deviation. This procedure yields the complete neural harmonization loss:

$$\mathcal{L}_{\text{Harmonization}} = \lambda_1 \sum_i^N ||(z \circ \mathcal{P}_i \circ g(f_\theta, x))^+ - (z \circ \mathcal{P}_i(\phi))^+||_2 \tag{1}$$

$$+ \mathcal{L}_{CCE}(f_\theta, x, y) + \lambda_2 \sum_i \theta_i^2 \tag{2}$$

Following the original *neural harmonizer* implementation [7] and training recipe, we trained six DNNs on ImageNet and human feature importance maps from the *ClickMe* game [61]: VGG16 [25], LeViT [62], ResNetV2-50 [63], EfficientNet_b0 [22], ConvNext [32], and MaxViT [64]. Each model was trained with Tensorflow 2.0 on 8 V4 TPU cores with all of the images in the ImageNet training set, and *ClickMe* human feature importance maps for the 200,000 images which were annotated. Images and feature importance maps were augmented with mixup [65], random left-right flips, and random crops during training. Only the object recognition loss was computed for images without human feature importance maps. Model weights were optimized with stochastic gradient descent and momentum, batches of 512 images, label smoothing [66], a learning rate of 0.3, and a learning rate schedule that began with a 5-epoch warm-up period followed by a cosine decay over 90 epochs at steps 30, 50 and 80.

**Neural fitting.** We evaluated the neural fit of each model by computing their fit separately for each IT recording, using the Brain-Score evaluation method [11]. This method involved fitting image-evoked activities from a layer of a deep neural network (DNN) to the corresponding neural responses using the partial least squares regression algorithm from Scikit-Learn.

To implement the Brain-Score evaluation method on the spatially-resolved recordings we investigate here, we first split each image shown to each animal into an equal number of patches as there were fixation locations in an image. Each image patch captured the receptive field of recorded neurons, and in total, there were $4,046$ image patches for ML and $4,046$ image patches for PL in Monkey 1, and $1,134$ image patches for ML in Monkey 2. We measured DNN neuronal fits as the Spearman correlations between model predictions and true neural responses for each patch of an image held out of training divided by each neuron's noise ceiling. We then stored the median Spearman correlation over neurons and repeated the training/testing procedure to get the mean correlation across all images viewed by a monkey. Separate fitting procedures were performed for every layer of activities in a model, and we report a DNN's Brain-Score as its best possible fit across layers.

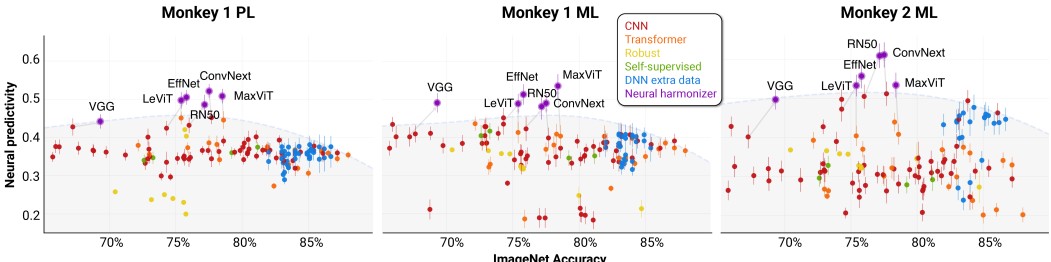

Figure 3: **DNNs trained on ImageNet face a trade-off between achieving high object recognition accuracy and predicting responses to natural images in IT.**. We measured the accuracy of 135 DNNs at predicting image-evoked responses from neurons in posterior lateral (PL) and medial-lateral (ML) areas of IT [6] by computing the neural predictivy of each model with the Brain-Score evaluation method [11]. DNN neural predictivity is progressively worsening as models improve on ImageNet. This problem can be partially fixed by training DNNs with the *neural harmonizer*, which aligns their learned object representations with humans. Error bars denote 95% bootstrapped confidence intervals.

## 3   Results

**Task Optimization is insufficient for reverse-engineering IT.**   Ever since it was found that DNNs optimized for object recognition produce accurate predictions of IT neural responses to images, it was suggested that prediction accuracy would continue improve alongside DNN performance on ImageNet [10–12]. Is this the case with today's DNNs that rival or exceed the performance of human object recognition?

To answer this question, we leverage recordings of neuronal responses to high-resolution natural images in medial (ML) and posterior lateral (PL) IT (see Methods and Fig. 2). These images fall within the same statistical distribution as ImageNet images (Fig. S1), ensuring that our findings are not influenced by distributional shifts that are a well-known problem faced by computer vision models [67]. Image-evoked neural responses were spatially mapped, enabling insights into the visual features driving the responses of IT neurons and DNNs. Moreover, while these regions were localized according to their selectivity to face stimuli, they were also noted to respond to non-face stimuli [6].

Across 135 different DNNs, representing the variety of approaches used today in computer vision, we found that DNNs pretrained on ImageNet have become progressively less accurate at predicting ML and PL responses in two separate Monkeys (Fig. 3). For instance, `ConvNext tiny`, which achieved 74.3% accuracy on ImageNet, is as accurate in predicting neural responses to images as the `ResNetv2-152x4`, which reached 84.9% accuracy on ImageNet. Moreover, training routines that have been suggested to be more biologically plausible or yield representations that are closer to biological vision, such as training with self-supervision [68] or for adversarial robustness [69], made no difference. All DNNs trained on internet data faced a pareto-front that bounded their accuracy in predicting IT responses as a function of their ImageNet accuracy.

**DNNs need biologically-aligned training routines**   There are at least two potential reasons why ImageNet-trained DNNs face a trade-off between object recognition and neural prediction accuracy: (*i*) DNN data diets and training routines work well for ImageNet but lead them to learn features

that are misaligned with biological vision [7, 17–19] or (*ii*) their architectures are a poor match to visual cortex [20]. To test this first possibility, we turned to the *neural harmonizer*. Given prior work demonstrating that aligning DNNs with human perception using the *neural harmonizer* significantly improved their ability to predict human behavior, we reasoned that harmonization might similarly improve DNN explanations of neural data by forcing them to rely on human-like visual features. Indeed, we found that harmonized DNNs (hDNNs) were significantly more accurate at predicting image-evoked responses in ML ($T(5) = 5.30, p < 0.01$) and PL ($T(5) = 6.11, p < 0.001$) neurons of Monkey 1 and ML ($T(5) = 6.89, p < 0.001$) neurons of Monkey 2 (reported as the average of $T$ score from independent samples $t-$tests comparing the predictivity of each hDNN to the DNN with the highest neural-predictivty). The success of hDNNs indicates that the architectural mismatch between DNNs and the visual cortex is far less of a problem for modeling the visual cortex than the training routines and data diets that are being used today to achieve high accuracy on ImageNet. Behavior – even from humans – is an important constraint on DNNs that is needed to build more accurate models of primate visual cortex.

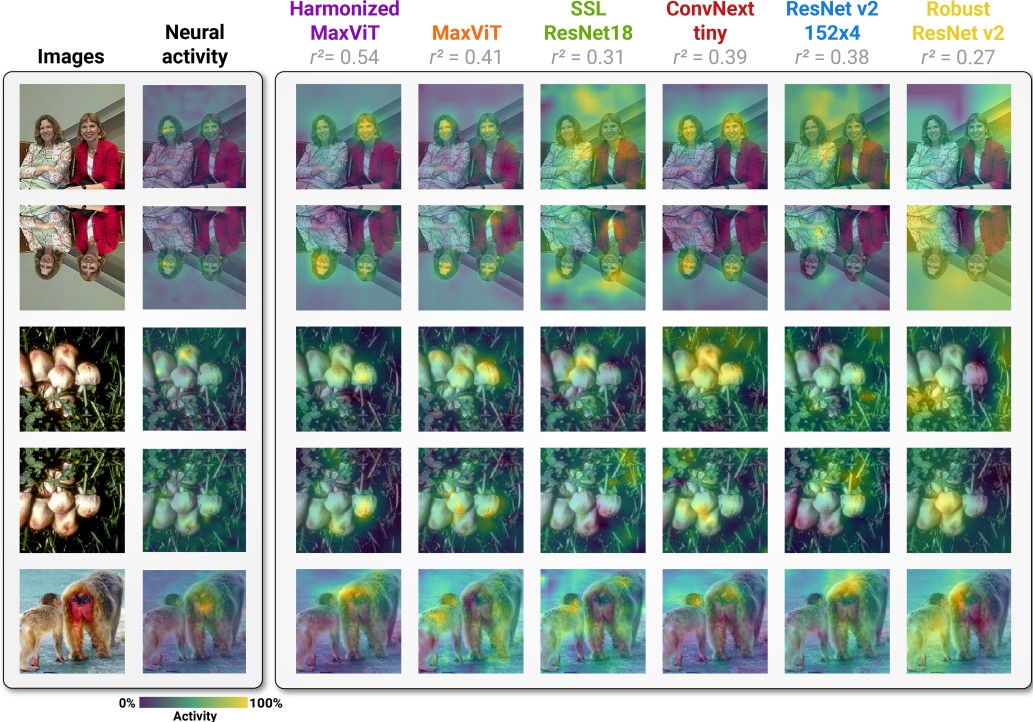

Figure 4: **DNNs optimized for object recognition on ImageNet rely on different features than those encoded by neurons in primate inferior temporal (IT) cortex.**. The activity of PL IT neurons is plotted next to the predicted activity of a model representing each class of DNNs: harmonized DNNs (hDNNs), visual transformers, self-supervised DNNs, convolutional neural networks, DNNs trained on more data than ImageNet, and adversarially robust DNNs.

**Spatially-mapped neural responses reveal a feature mismatch between IT and DNNs**  The IT recordings used in the Brain-Score benchmark all utilized the same paradigm, in which hundreds or thousands of images were shown to multiple animals during passive viewing [14]. In contrast, the recordings we rely on involve many fewer unique images, but neural responses for each image are spatially mapped. This spatial mapping reveals what types of features are driving neural responses and makes it easier to understand the successes and failures of DNNs in explaining these data [6].

Although the neurons in ML and PL were located based on their selectivity to faces, the spatial maps of their responses revealed a much more complex response profile. IT neurons in both animals were strongly driven by faces, non-face objects, and contextual cues associated with faces [6]. In contrast, the best-fitting units of DNNs with state-of-the-art accuracy on ImageNet, like the `ResNetv2-152x4` (Fig. 4), responded strongly to background features. This problem was shared by DNNs trained

on internet data but with routines that have been considered more biologically plausible, like self-supervised learning (e.g., `SSL ResNet-18`), or routines that yield perceptual robustness that is closer to humans, like adversarial training (e.g., `Robust ResNetv2-50`). hDNNs like the `harmonized MaxViT`, on the other hand, were, in general, reliably predictive of what parts of images elicited the most activity from IT neurons.

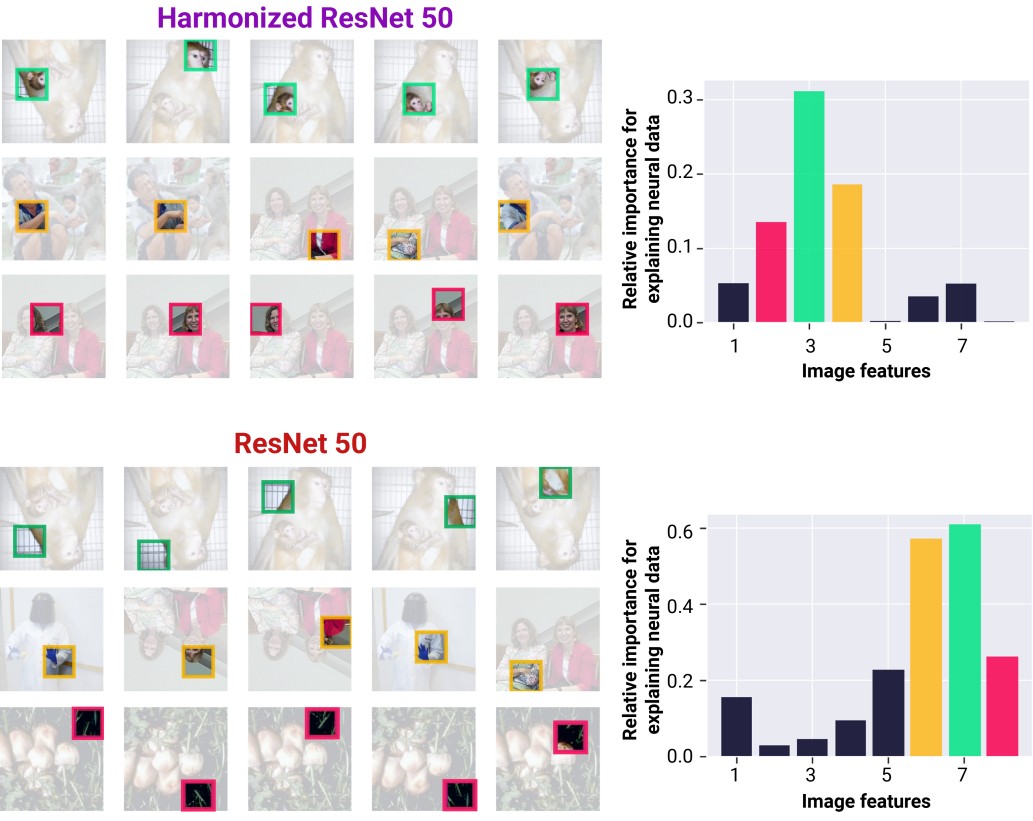

Figure 5: **Predictions of the visual features drive neuronal responses in PL of Monkey 1 from a harmonized ResNet-50 and a standard ResNet-50.** Image patches for each model depict what the important features are across all images shown to the animal. The relative importance of each feature for predicting the recordings is color-coded in the bar chart on the right.

**hDNNs generate testable predictions for visual neuroscience**   The surprising effectiveness of DNNs pre-trained on object recognition for predicting IT responses to images was a significant finding in computational neuroscience, which raised the tantalizing possibility that these models could form the basis of neuroprosthetics, and reduce the need for animal experiments. However, a persistent problem with DNNs since their inception is their lack of interpretability, meaning that using DNNs for systems identification effectively swaps one black box (the visual system) with another (a DNN) without getting the field any closer to understanding how vision works. Recent strides in the field of explainable artificial intelligence (XAI) have begun to alleviate this problem. For instance, it is now possible to reliably locate and characterize the features in datasets, like ImageNet, that drive patterns of model behavior using concept recursive activation factorization (CRAFT [70]). When paired with accurate models of IT and recordings that reveal locations in images that elicit neuronal responses, CRAFT can generate testable predictions of *what* those features are.

We used CRAFT to extract features from harmonized and regular ResNet-50s trained on ImageNet that explain their predictions of neural activity in PL of Monkey 1 (Fig. 5). To do this, we decomposed a DNNs predicted activities for image patches into the most important and distinct features using non-negative matrix factorization, then scaned over every single image patch to find those which most strongly explained each discovered feature. While the `harmonized ResNet-50` predicted that parts of faces, arms, and heads explained the majority of variance in IT, the less accurate and unharmonized

`ResNet-50` predicted that the background, oriented-edges, and a feature combining head, body, arm, and hand parts were most important for IT. Thus, hDNNs not only break the trade-off between ImageNet and neural prediction accuracy of ImageNet-trained DNNs, they can also generate testable hypotheses on the relative importance of different features for visual system neurons.

## 4  Related work

**The scaling laws of deep learning**    The immense gains of DNNs in computer vision [13] and natural language processing [71] over recent years share a common theme: they have relied on unprecedented computational resources to train models with billions of parameters on internet-scale datasets. These benefits of scale have been studied over the past several years across a number of domains and often yield predictable improvements on standard internet benchmarks [71, 72]. Surprisingly, scale has also caused models to exhibit human-like behavior in psychophysics experiments [13, 57, 73, 74]. In other words, many aspects of primate behavior *are* captured progressively better by DNNs as they scale-up and improve in accuracy on benchmarks like ImageNet.

**Scale is sufficient for modeling internet benchmarks, but not primate intelligence**    In parallel, there's a growing body of research suggesting that large-scale DNNs are becoming less aligned with human perception in multiple ways. For instance, the representations and visual decision-making behavior of DNNs are inconsistent with humans, and this problem has worsened as they have improved on ImageNet [7]. DNNs are also becoming less capable at predicting perceptual similarity ratings of humans, and they remain vulnerable [75] to so-called "adversarial attacks." Our work adds to this body of research, indicating that current scaling laws are incompatible with modeling primate behavior *and* poorly suited for explaining the neural computations that shape it.

**Aligning DNNs with primate vision**    There have been a number of methods proposed for aligning DNNs with primate vision beyond the *neural harmonizer* that we leverage here [7]. It was found that co-training DNNs for object recognition and minimizing representational dissimilarity improved the adversarial robustness of a recurrent DNN [76]. Others have shown that similar forms of representational alignment improve few-shot learning and predictions of human semantic judgments in DNNs [77, 78]. The successes of these behavioral and representational alignment methods highlight the real limitations of current DNN training routines and data diets for generating artificially intelligent systems that act like biological ones.

**Biological learning routines**    There have been many efforts to align the object functions, learning rules, and data diets of DNNs more closely with biological systems. It was found that DNNs trained with self-supervision on ImageNet instead of supervised recognition achieve similar accuracy in predicting V1, V4, and IT responses to images [68]. DNNs trained with self-supervised learning on head-mounted camera video from infants instead of ImageNet were approximately as accurate at predicting neural data as DNNs trained on Imagenet but needed far less data to do this [68]. Others have found that adversarial training of DNNs yields models that are more accurate at predicting neural responses to images [79] and have similar tolerance to adversarial attacks as neurons in IT [69].

## 5  Discussion

**A revised approach for reverse engineering visual cortex**    Biological data is expensive to gather and often noisy. This makes the prospect of accurately modeling the responses of IT to complex stimuli especially daunting. DNNs optimized for object recognition represented a potential solution for this problem: pretraining on internet datasets alleviated training issues associated with small-scale neural data, and the architectures and training routines of accurate DNNs could offer insights into the circuits that underlie visual perception as well as the developmental principles that shape those circuits. Our findings suggest that while DNNs still hold great potential for predicting image-evoked responses from IT neurons, new training routines and data diets are necessary for continued improvement.

The *neural harmonizer* is a partial solution to the problems that DNNs face in modeling primate IT. The success of hDNNs in breaking the pareto-front faced by 135 different DNNs indicates that significant aspects of primate perception and the neural circuits that shape it cannot be divined from internet data alone. We believe that the *neural harmonizer* and learning constraints provided by

large-scale human behavior data is only a short-term solution to this problem, and that if DNNs were able to learn about the visual world more like primates do, they would be even better at predicting neural data. In support of this goal, we release our code and data at `https://serre-lab.github.io/neural_harmonizer/`.

**Limitations.** One limitation of our work is that we use the responses of neurons with face-selectivity to replicate and understand the trade-off between ImageNet accuracy and neural predictivity faced by DNNs. As faces (and even humans) are not a category in ImageNet, it is possible that this dataset [6] could bias our results in ways that are difficult to predict[†]. However, we find the same ImageNet accuracy and neural predictivity trade-off on this dataset (Fig. 3) as we did on the three recordings of object selective neurons hosted on the Brain-Score website (Fig. 1), indicating that DNNs face similar issues in predicting each set of recordings. Moreover, the neural harmonizer was successful in breaking this trade-off, even though it involves ImageNet training using human feature importance maps for objects. Finally, it was noted in the original paper where our recordings came from that the neurons, while localized based on their face-selectivity, responded to non-face stimuli as well [6]. In summary, our work reliably demonstrates that new paradigms are needed to advance DNNs as models of IT, and spatially-resolved recordings such as those used in this work support this goal.

Another limitation of our work is that while we found that hDNNs are significantly more predictive of IT neuron responses than any other DNN, they still explain only 50-60% of the variance in neuronal activity. One straightforward way of doing better is by expanding the dataset of human feature importance maps we used for harmonization from annotations for approximately 200,000 images to the entire 1.2M ImageNet dataset.

**Broader impacts.** By building better models of primate IT, we are taking significant steps toward the reducing the reliance of visual neuroscience on animal models for experimentation, supporting the development of neuroprosthetic devices that resolve visual dysfunctions, and providing vision scientists with a richer understanding of how IT works. Our findings also highlight a main limitation of the scaling laws that are guiding progress throughout artificial intelligence today: scale is not sufficient for explaining biological intelligence.

## Acknowledgments and Disclosure of Funding

This work was supported by ONR (N00014-19-1-2029), NSF (IIS-1912280 and EAR-1925481), DARPA (D19AC00015), NIH/NINDS (R21 NS 112743), and the ANR-3IA Artificial and Natural Intelligence Toulouse Institute (ANR-19-PI3A-0004). Additional support provided by the Carney Institute for Brain Science and the Center for Computation and Visualization (CCV). We acknowledge the Cloud TPU hardware resources that Google made available via the TensorFlow Research Cloud (TFRC) program as well as computing hardware supported by NIH Office of the Director grant S10OD025181.

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

# A Datasets

**Distribution shifts**   One possible explanation for the trade-off we observed between DNN accuracy on ImageNet vs. predictions of neural data is that there was a distributional shift between the images used in the two evaluations. To test this possibility, we extracted activity vectors for a large sample of images from ImageNet and all images from the standard Brain-Score benchmark dataset for IT [14] and the images we used in our experiments [6]. For ImageNet, we used all images in the validation set that also had human annotations from *ClickMe*.

We measured distribution shifts by measuring the accuracy of a linear classifier in discriminating between ImageNet and either the standard Brain-Score benchmark dataset for IT [14] or the images we used in our experiments [6]. We began by taking equal-sized random samples of ImageNet image encodings and encodings from either the Brain-Score dataset (28 of 191 images) or our dataset (all 28 images). We then trained a linear SVM using scikit-learn and performed leave-one-out cross-validation on either set of samples. Finally, we repeated this procedure 1,000 times, using a new random sample of ImageNet encodings each time, and measured the average accuracy for discriminating between ImageNet and either neural data image set. We found that images used in the standard Brain-Score benchmark for IT [14] are significantly further out-of-distribution from ImageNet than the images that we used in the main text [6] ($T(999) = 98.12$, $p < 0.001$, Fig. S1).

Figure S1: **Stimuli from [14], but not [6], are out-of-distribution from ImageNet.** Penultimate-layer activities from a ResNetv2-50 trained on ImageNet were extracted for 1916 images from the validation set of ImageNet, 191 images from [14], and 28 images from [6]. A principle components analysis decomposed those activities into a 2-Dimensional projection, which demonstrated a distribution shift for stimuli from [14], but not [6]. Indeed, while stimuli from [14] could be reliably discriminated from ImageNet (98.7 acc, $T(999) = 98.12$, $p < 0.001$), images from [6] were not discriminable from ImageNet (56.5 acc, $T(999) = 98.12$, $n.s$
.

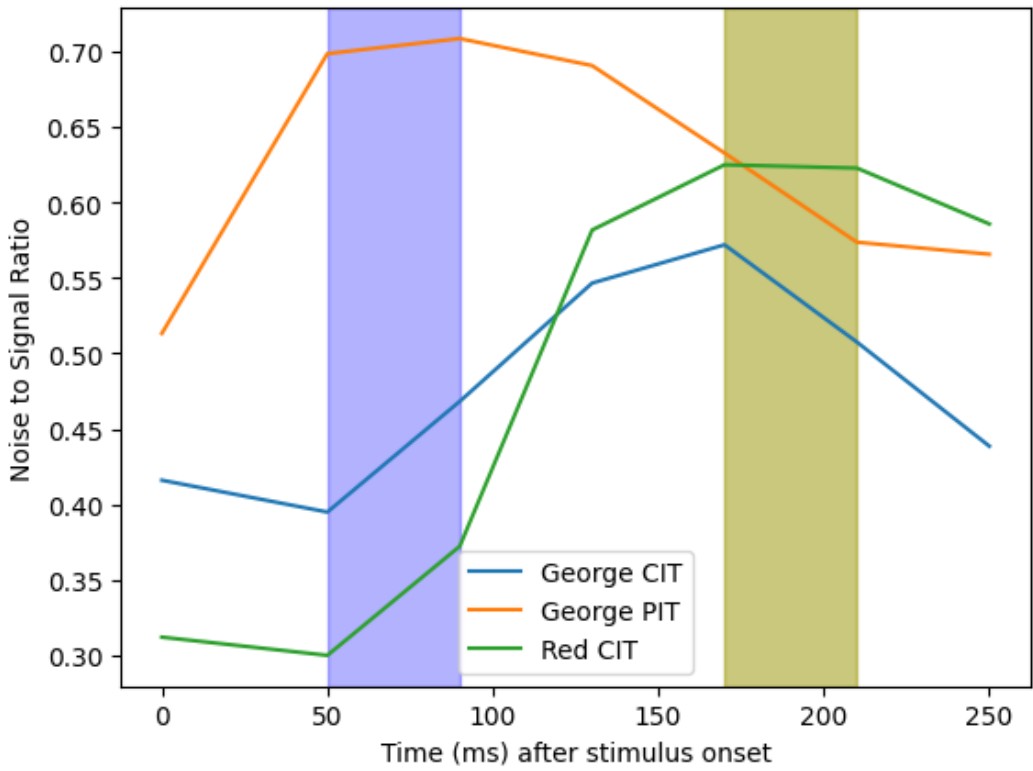

Figure S2: **Noise ceilings at every 40ms bin of activity from 0ms - 250ms.** We binned every 40ms of neural responses following the original procedure for processing this data [6]

## B   Neural data analysis

We measured the maximum explainable variance, or noise ceiling, in our neural data recordings in every 40ms bin of activity from 0ms - 250ms. We binned every 40ms by following the original procedure for processing this data [6]. Because images were not repeatedly shown to the same population receptive field, we computed each neuron's noise ceiling for an image as the highest correlation between it and another neuron for that image. We then focused our analyses in the main text on the time bin where each neurons in each animal/brain area achieved the highest average noise ceiling (Fig. S2).

## C   Models

**DNN Model Zoo**   We comprehensively evaluated the ability of a myriad of DNNs from the TIMM toolbox. These DNNs, available under the Apache 2.0 license, are intended for non-commercial research purposes. The complete list of DNNs we evaluated can be found in Table $S1$.

## D   *Neural Harmonizer* Training

We relied on the original neural harmonizer recipe [7] to train and harmonize 14 DNNs for object recognition on the ImageNet [9] dataset. For each model, we searched for settings of the regularization terms $\lambda_1$ and $\lambda_2$, which control the relative importance of losses for object recognition and human feature alignment during training, that maximized accuracy on the ImageNet validation set. In total, we trained a `VGG16`, `ResNet50_v2`, `ViT_b16`, `EfficientNet_b0`, `ConvNext Tiny`, and `MaxViT Tiny` (Table. S2). We did not attempt to apply the neural harmonizer to models which relied on pretraining with datasets other than ImageNet because the *ClickMe* feature importance dataset we used only contained annotations on a subset of images in ImageNet. Thus, we expect that

| Architecture | Model | Versions |
|---|---|---|
|  | VGG | 4 |
|  | ResNet | 24 |
|  | EfficientNet | 8 |
|  | ConvNext | 10 |
|  | MobileNet | 6 |
|  | Inception | 7 |
|  | DenseNet | 4 |
|  | RegNet | 3 |
|  | Xception | 4 |
| CNN | MixNet | 4 |
|  | DarkNet | 2 |
|  | NFNet | 2 |
|  | TinyNet | 4 |
|  | LCNet | 2 |
|  | DLA | 2 |
|  | MnasNet | 2 |
|  | Resnext101 | 6 |
|  | GhostNet | 2 |
|  | General ViT | 21 |
|  | leViT | 4 |
| ViT | MaxViT | 7 |
|  | DeiT | 4 |
| Hybrid | CoAtNet | 5 |

Table S1: **Our DNN model zoo, taken from the TIMM library.**

| Model | Accuracy | Human Alignment | Note |
|---|---|---|---|
| VGG | 69.3 | 61.5 | $\lambda = 2$ |
| ResNet 50 | 77.17 | 45.0 | $\lambda = 2$ |
| EfficientNet B0 | 77.51 | 52.3 | $\lambda = 20$ |
| ViT B16 | 75.7 | 72.6 | $\lambda = 5$ |
| ConvNext Tiny | 75.9 | 73.2 | $\lambda = 1$ |
| MaxViT Tiny | 78.6 | 45.3 | $\lambda = 1$ |

Table S2: **DNNs trained with the *neural harmonizer*.**

more work is needed for expanding *ClickMe* to larger-than-ImageNet scale data before these DNNs can be harmonized.

# E   Extended results

**Cross validation**   One limitation of the evaluation approach used in the original Brain-Score work [11] is that model selection is done on the same dataset as testing (*i.e.*, there is no partitioning of the data into separate training, validation, and test sets). Thus, it is possible that the results we describe in the main text are a byproduct of "double dipping." To test for this possibility, we repeated our experiments using separate training, validation, and test splits of the data. For each round of cross validation in this procedure, we held out one image for testing. We also held out half of the training data for selecting the model layer that best predicted neural responses (validation). Our results from this approach strongly correlated with the ones reported in our original submission and our overall conclusions remain the same (Fig. S3; Monkey 1 ML $\rho = 0.98$, $p < 0.001$; Monkey 2 ML $\rho = 0.97$, $p < 0.001$); and Monkey 2 PL $\rho = 0.98$, $p < 0.001$).

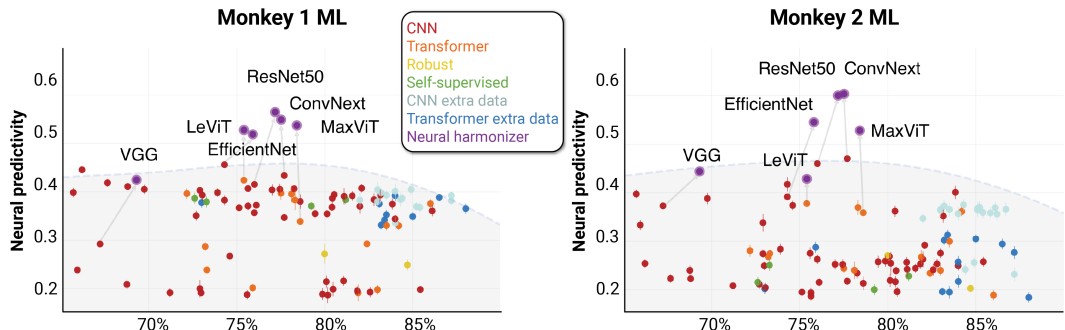

Figure S3: **DNN neural prediction accuracy after evaluating on a test set held out of training.** Unlike the original approach introduced in Brain-Score [11], here we train, validate, and test models on separate partitions of the dataset. DNNs trained and evaluated in this way follow the trends described in the main text.

| Task | ML | PL |
|---|---|---|
| Autoencoding | 0.005 | -0.009 |
| Class object | 0.091 | 0.075 |
| Curvature | 0.019 | -0.009 |
| Denoising | -0.014 | 0.013 |
| Edge occlusion | 0.029 | -0.0002 |
| Edge texture | 0.025 | 0.014 |
| Egomotion | 0.002 | -0.018 |
| Fixated pose | 0.118 | 0.101 |
| Jigsaw | 0.099 | 0.097 |
| Keypoints 2D | 0.024 | 0.015 |
| Keypoints 3D | -0.057 | -0.006 |
| Nonfixated pose | 0.164 | 0.086 |
| Normal | 0.065 | 0.056 |
| Reshading | 0.019 | 0.001 |
| Room layout | 0.131 | 0.087 |
| Segment semantic | -0.011 | 0.029 |
| Segment unsupervised 25D | 0.041 | 0.008 |
| Segment unsupervised 2D | 0.052 | 0.043 |
| Vanishing point | 0.046 | 0.076 |

Table S3: **Neural predictions for ML and PL in Monkey 1 using Taskonomy-trained DNNs.**

**Alternative objective functions and datasets**    To better understand how ImageNet *per se* affected DNNs' abilities to predict neural responses, we evaluated a zoo of them trained on the Taskonomy dataset and task set [80] and also the Ecoset [81] naturalistic object categorization dataset and task. We took 19 DNNs pretrained on each task in the Taskonomy, and 4 DNNs trained for classification on ecoset, and applied the standard Brain-Score fitting procedure that we describe in our original submission Methods to derive prediction accuracy scores for each model. These models were far less effective at explaining neural activity than any of the ImageNet-trained models we provided in the main text (Tables S3 and S4). To summarize, our human behavior-aligned harmonized DNNs are more accurate at predicting neural responses than other DNN we tested, regardless of the dataset or task that was used to train them.

| Model | Best time bin | Score |
|---|---|---|
| Alexnet | 90 | 0.046 |
| ResNet50 | 130 | 0.111 |
| VGG16 | 130 | 0.086 |
| Inception | 210 | 0.109 |

Table S4: **Neural predictions for ML in Monkey 1 using Ecoset-trained DNNs.**

| Model | Neural | ImageNet |
|---|---|---|
| VGG baseline | 0.523 | 67.23 |
| VGG harmonized | **0.524** | **67.369** |
| ResNet50v2 baseline | 0.501 | 76 |
| ResNet50v2 harmonized | **0.502** | **77.17** |
| ConvNext baseline | 0.519 | 74.3 |
| ConvNext harmonized | **0.524** | **75.8** |
| MaxViT baseline | 0.541 | 78.6 |
| MaxViT harmonized | **0.543** | **78.7** |
| EfficientNetB0 baseline | 0.506 | 75.4 |
| EfficientNetB0 harmonized | **0.508** | **77.51** |
| LeViT baseline | 0.472 | 74.9 |
| LeViT harmonized | **0.473** | **75.4** |
| ViT baseline | 0.493 | 73.2 |
| ViT harmonized | **0.499** | **75.7** |

Table S5: **Performance of select DNNs on inferotemporal (IT) recordings from** `Brain-score.org` **[14].** For each DNN class, models with higher accuracy at predicting IT responses and ImageNet recognition are bolded.

**Performance on Brain-Score**    As a point of comparison, we tested a subset of our DNN on a standard inferotemporal (IT) recordings benchmark on `Brain-Score.org`. In each case, harmonization marginally improved performance over baseline (Table S5).

**Neural dynamics**    We also present modeling results on other time bins (with lower noise ceilings) than those considered in the main text (Figs. S4- S6). For ML IT in monkey 1 and monkey 2, we present 90ms-130ms, 130ms-170ms, and 210ms-250ms. For PL IT of Monkey 1, we present 130ms-170ms, 170ms-210ms, and 210ms-250ms, windows. In each case, we found a similar trade-off in these data as we did in those highlighted in the main text, as well as a significant improvement for harmonized vs. unharmonized DNNs. This similarity extends to the qualitative predictions made by each model on ML IT neurons (recall that we show the image-evoked activities of PL IT neurons in the main text, S6).

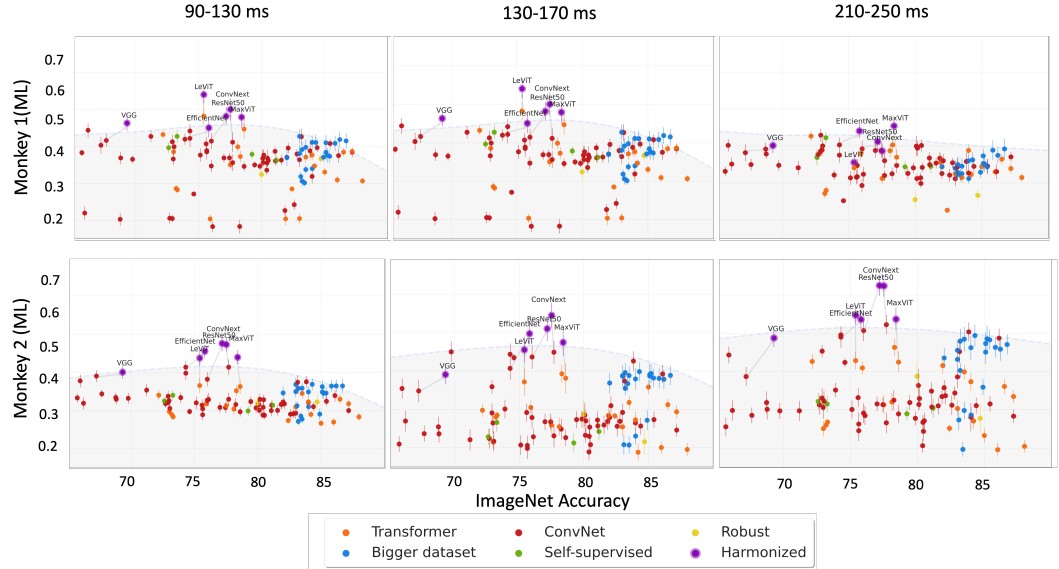

Figure S4: **Results using different time bins than in the main text for the same region (ML) for both monkeys.** The pattern of results for each time-bin are consistent with the main text, despite these data having lower noise ceilings.

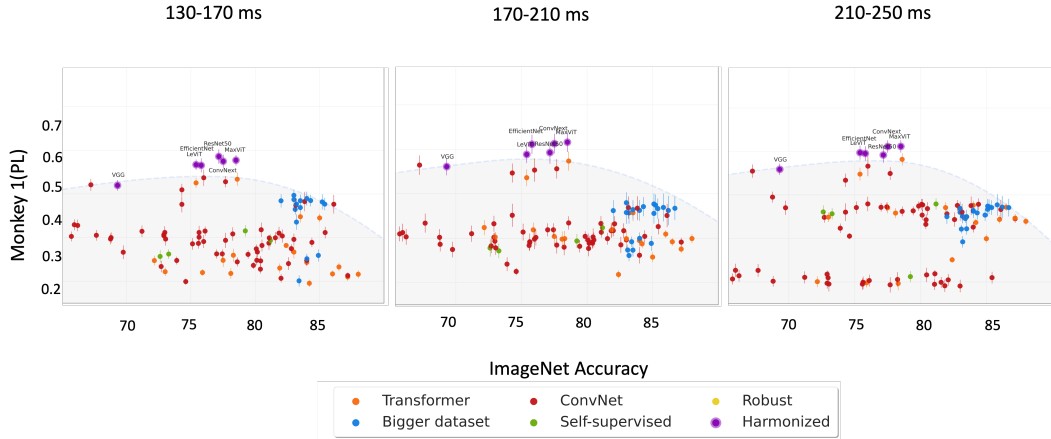

Figure S5: **Results using different time bins than in the main text for the IT PL in Monkey 1.**. The pattern of results for each time-bin are consistent with the main text, despite these data having lower noise ceilings.

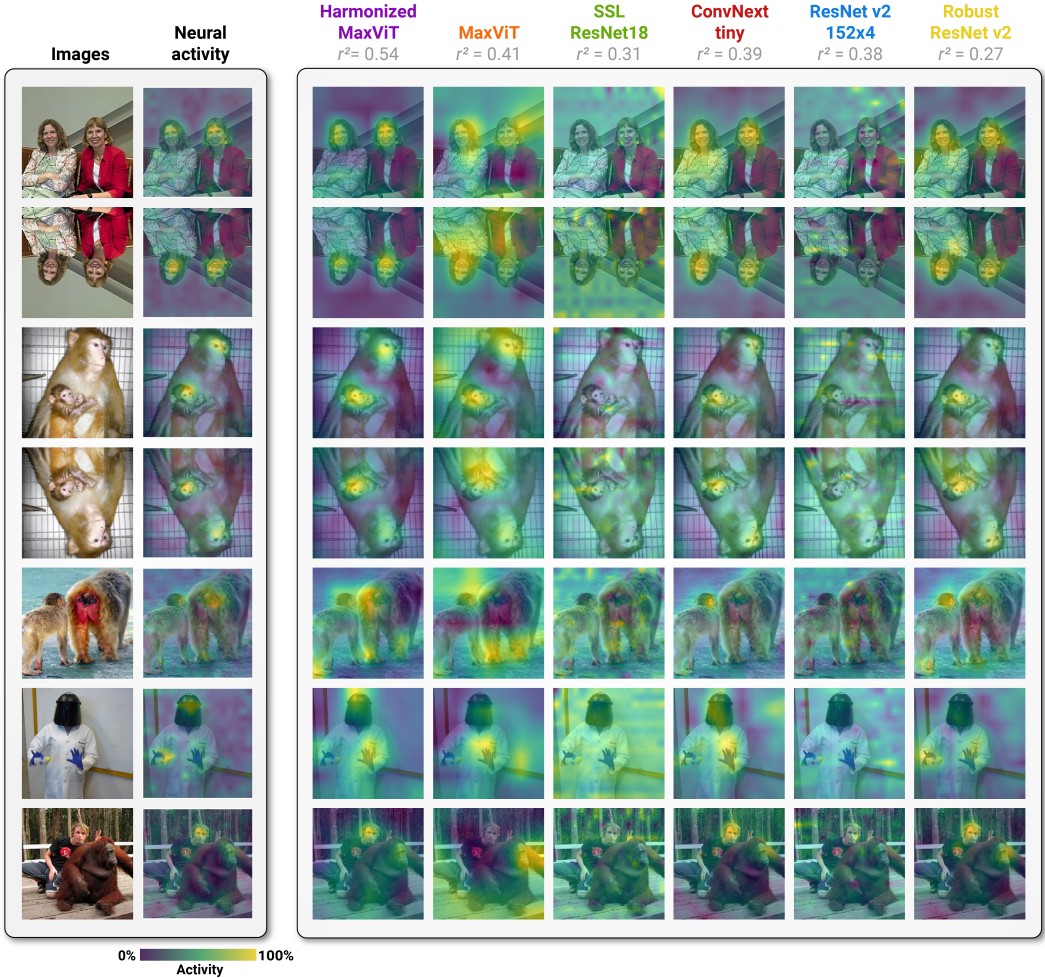

Figure S6: **DNNs optimized for object recognition on ImageNet rely on different features than those encoded by neurons in primate inferior temporal (IT) cortex.** The activity of ML IT neurons of Money 1 is plotted next to the predicted activity of a model representing each class of DNNs: harmonized DNNs (hDNNs), visual transformers, self-supervised DNNs, convolutional neural networks, DNNs trained on more data than ImageNet, and adversarially robust DNNs.

