# OpenReview forum: "Performance-optimized deep neural networks are evolving into worse models of inferotemporal visual cortex"
_NeurIPS.cc/2023/Conference — NeurIPS 2023 poster_

### Official Review · Reviewer_itup · 2023-07-03

**Soundness:** 3 good
**Presentation:** 2 fair
**Contribution:** 3 good
**Rating:** 6
**Confidence:** 4

**Summary:**

The paper discusses the observation that improved imagenet classification performance is no long correlated with neuron prediction performance in macaque IT.

To validate this claim, the authors perform experiments where an image was moved across the visual field while the monkey maintained fixation. This process allowed the authors to examine the relative importance of the spatially distributed features in an image.

The authors further compared against a set of architectures from pytorch timm, including those trained on imagenet, self-supervised models, and those trained on internet-scale datasets. Both CNNs and ViT-like models were tested.

The authors find that aligning the gradients of a model with human data improved the prediction performance of DNNs for IT neurons.

In the supplementary, the authors show that their stimuli is better aligned to imagenet, and thus their results cannot be explained by the dataset misalignment.

**Strengths:**

The paper is interesting and timely.

With the widespread availability of datasets with hundreds of millions to billions of images (YFCC, Conceptual Captions, Datacomp-1B, LAION-2B), models have repeatedly achieved higher performance on zero-shot imagenet. Researchers building encoding models have increasingly embraced these models as backbones.

The paper shows that imagenet performance for a variety of different models is no longer strictly correlated with brain predictive performance.

The idea of shifting images while maintaining eye fixation is an interesting way to find biological importance of different features.

**Weaknesses:**

* On DNN fitting
  * The paper provides few details on how exactly the images were presented to DNNs. For the monkey presentation, fixation was maintained and the images were shifted. Line 141 onwards provides an explanation, however it is very unclear how exactly you perform this step.
* On the models used
  * I find the division of models into CNN/Transformer/Robust/Self-supervised/DNN extra data to be confusing. For example CLIP models would presumably fall under DNN extra data. However CLIP models have both CNN and ViT variants, how are models classified?
  * I suggest the authors use a mutually exclusive division by architecture (CNN/ViT/MLP-mixer etc), loss or dataset (supervised ImageNet, supervised large dataset e.g. CLIP on the billion+ images, self-supervised using patch/rotation/other prefix tasks)
  * This applies to Figure 1 and Figure 3, as well as discussion within the text.
* On Neural harmonizer
  * The section starting from Line 116 provide very sparse details for how they actually train their neural harmonizer
  * What is $g$? Is this the gradient operator? What is the gradient with respect to? Is it to the mean of the neurons in IT?
  * Are you performing a second-order gradient optimization?

* On the feature importance figures
  * How exactly are you plotting feature importance for DNNs? What approach are you using? GradCAM? In text you mention CRAFT in line 215, however it is not clear if this is the approach you actually use for the visualizations
  * Presumably you don't use CRAFT for the neural harmonizer? If so, please clarify what you use for the visualization and the neural harmonizer.

* On imagenet accuracy
  * Was imagenet accuracy top-1 accuracy? Was this used dataset ImageNet-1k? ImageNet-21k? ImageNetv2?
  * How was accuracy for contrastive models computed? Were they zero-shot probes in the style of "a photo of a x" as in the CLIP paper?

* Minor typos and grammatical errors
  * Line 58 -> "tjat"
  * Line 119 -> "let $P$ be a function that a multi-scale Gaussian pyramid of a human feature importance map ...", this line is super confusing.
  * Supplementary code "realising" -> "releasing"

* On related work
  * I recommend the authors discuss at least one of the following papers:
  * [1] which show how most DNNs trained on visual-language contrastive losses cannot perform compositional reasoning
  * [2] which discusses how recognition and retrieval tasks, and the dataset used to train these tasks lead to compositional features not being emphasized in DNNs
  * [3] which discusses how DNN initializations yield representational gaps between vision and concepts


The paper would be strengthened with additional details and clarifications.

[1] Thrush, Tristan, et al. "Winoground: Probing vision and language models for visio-linguistic compositionality." Proceedings of the IEEE/CVF Conference on Computer Vision and Pattern Recognition. 2022.

[2] Yuksekgonul, Mert, et al. "When and Why Vision-Language Models Behave like Bags-Of-Words, and What to Do About It?." The Eleventh International Conference on Learning Representations. 2022.

[3] Liang, Victor Weixin, et al. "Mind the gap: Understanding the modality gap in multi-modal contrastive representation learning." Advances in Neural Information Processing Systems 35 (2022): 17612-17625.

**Questions:**

See weaknesses section for other questions.

For figure 4, did you actually present upside down images to the monkeys? Or were they only used for the training of the neural harmonizer?

**Limitations:**

The authors largely address the limitations of their paper. However the authors could potentially emphasize that their approach applies to data collected from electrophysiology from macaque monkeys. Recent work on human fMRI data suggests that contrastive optimized DNNs (CLIP) are better encoding models of the human visual cortex than fully supervised models (imagenet).

[1] Conwell, Colin, et al. "What can 5.17 billion regression fits tell us about artificial models of the human visual system?." SVRHM 2021 Workshop@ NeurIPS. 2021.

---

> ### Author Rebuttal · Authors · 2023-08-09
>
> > The paper provides few details on how exactly the images were presented to DNNs. For the monkey presentation, fixation was maintained and the images were shifted. Line 141 onwards provides an explanation, however it is very unclear how exactly you perform this step.
>
> We apologize for the lack of clarity. We presented every image shown to a monkey to a DNN and then extracted the feature map at a given layer. Then, a feature map patch was extracted at each of the monkey’s receptive field locations. These feature map patches were used to determine neural predictivity. We will revise our description in the main text with these details. We also included a link to our code with an implementation of this procedure on line 276.
>
> > I find the division of models into CNN/Transformer/Robust/Self-supervised/DNN extra data to be confusing. For example CLIP models would presumably fall under DNN extra data. However CLIP models have both CNN and ViT variants, how are models classified?
>
> Thank you for the suggestion! We have split the DNN extra data category into CNN extra data and Transformer extra data (Rebuttal Figs A and B). CLIP-ResNet falls into the former group and CLIP-Transformer falls into the latter group. We hesitate adding additional divisions as it will clutter the figures, but please take a look at our revised figures and let us know what you think.
>
> > The section starting from Line 116 provide very sparse details for how they actually train their neural harmonizer. What is g? What is the gradient with respect to? Are you performing a second-order gradient optimization?
>
> We apologize for the oversight! Here, g(.) refers to any attribution method (heatmap method), and indeed, we utilize gradients. The loss is first differentiated with respect to the input, resulting in df(x)/dx (yielding a heatmap). Next, we compute the derivative of the difference between this heatmap and the desired heatmap with respect to our model's weights. This involves a mixed partial derivative process. We'd like to clarify that even with ReLU activations, the gradient of the loss is not zero (a trivial case to comprehend is calculating the closed-form of the gradient of the loss for a dense ReLU network). Therefore, the optimization occurs just as with any other training process. We will incorporate this explanation into our revision.
>
> > How exactly are you plotting feature importance for DNNs? What approach are you using? GradCAM? In text you mention CRAFT in line 215, however it is not clear if this is the approach you actually use for the visualizations
>
> We plotted predicted and actual neural responses at every receptive field location in an image (see Figure 4 caption). These are not attribution maps. We will clarify this in the main text. We also applied CRAFT to harmonized and non-harmonized ResNet50 models. CRAFT provides a complementary insight: what are the actual features (as opposed to an activity heatmap) that each model thinks are driving IT responses? CRAFT shows that harmonized models predict that face features are more important for explaining neural responses, whereas nonharmonized models are less selective for face features.
>
> > Was imagenet accuracy top-1 accuracy?
>
> We plotted DNN top-1 accuracy on ILSVRC12/ImageNet-1k. Some of the models in our zoo were trained on data beyond just ImageNet (CNN/Transformer extra data, original submission Figs. 3/4 and rebuttal figs A/B). We will clarify in the revision.
>
> > How was accuracy for contrastive models computed? Were they zero-shot probes in the style of "a photo of a x" as in the CLIP paper?
>
> As noted in our Methods section, nearly all models in our zoo were taken from TIMM, and the rest are from the Neural Harmonizer. The CLIP models were taken from TIMM, where they were finetuned on ImageNet-1k.
>
> > Minor typos and grammatical errors
>
> Thank you, these were fixed.
>
> >On related work, I recommend the authors discuss at least one of the following papers: [1] which show how most DNNs trained on visual-language contrastive losses cannot perform compositional reasoning, [2] which discusses how recognition and retrieval tasks, and the dataset used to train these tasks lead to compositional features not being emphasized in DNNs, [3] which discusses how DNN initializations yield representational gaps between vision and concepts.
>
> Thank you for these references. We will include them in our discussion.
>
> > For figure 4, did you actually present upside down images to the monkeys?
>
> Yes the monkeys actually saw the images as presented in Figure 4. We will clarify this point in an expanded methods section in the appendix of our submission.
>
> > The authors largely address the limitations of their paper. However the authors could potentially emphasize that their approach applies to data collected from electrophysiology from macaque monkeys. Recent work on human fMRI data suggests that contrastive optimized DNNs (CLIP) are better encoding models of the human visual cortex than fully supervised models (imagenet).
>
> Thank you for this comment. As discussed in our main rebuttal, we will add a discussion of how model-based work on fMRI relates to our findings in our manuscript.

---

> > ### Comment · Reviewer_itup · 2023-08-11
> >
> > I thank the authors for the response, they have indeed clarified my questions. I have decided to retain my original score.

---

### Official Review · Reviewer_GM1r · 2023-07-06

**Soundness:** 4 excellent
**Presentation:** 4 excellent
**Contribution:** 3 good
**Rating:** 7
**Confidence:** 4

**Summary:**

This work address an important question in the construction of computational models of object recognition: the increase in performance of recent DNN models is not anymore accompanied (like in the past) by an increase in their ability to predict neural responses. This is a very relevant problem for the advancement of theoretical  neuroscience and its medical applications.

The Authors make extensive analysis of this detachment, using the Brain-Score metrics, and investigate two possible sources of this phenomenon, based respectively on the data and the architectures employed.
Crucially, they employed a dataset with realistic stimuli and with spatial information that can be connected with the neural activity.

Finally, the Authors introduce a technique called "neural harmonizer" that allows to partially align human and machine responses, and is suitable to generate interpretable hypothesis about the features that drive neural responses.

**Strengths:**

The paper is exceptionally well written and clear. It addresses a very relevant problem in computational neuroscience and its applications. It provides a convincing evidence that the neural harmonizer technique is effective in aligning human and DNNs responses, thus providing a base for important applications in prosthetics, reducing the need for animal experimentation. The originality comes from the combined action of using a new dataset, with spatial information, the usage of the neural harmonizer (that has already been used to align DNNs with human perceptual data) and extensive experimentation that gives strength to the conclusions.

**Weaknesses:**

I honestly can't find any that has not been already discussed in the Limitations section.

**Questions:**

Is it possible that, as a speculative question, the mismatch between the different features learned by DNNs and IT can be partially accounted for by the usage of backpropagation?

**Limitations:**

The limitations are critically discussed in the dedicated section, and the broader impacts are also addressed.

---

> ### Author Rebuttal · Authors · 2023-08-09
>
> > Is it possible that, as a speculative question, the mismatch between the different features learned by DNNs and IT can be partially accounted for by the usage of backpropagation?
>
> Fantastic question. As we wrote in our discussion, we believe that a wholesale revision of DNN training routines may be necessary for improving predictions of image-evoked neural responses in IT. In our paper we show that aligning DNNs with human behavior can partially achieve this goal, but we believe that great progress will be made in the future by identifying principles that could yield better predictions of neural activity without the need to co-train on human behavior. In our discussion we discuss opportunities for designing better datasets and objective functions for doing this, but more biologically-plausible learning algorithms are another approach that we will mention in our revision.

---

### Official Review · Reviewer_X5jQ · 2023-07-06

**Soundness:** 3 good
**Presentation:** 2 fair
**Contribution:** 3 good
**Rating:** 6
**Confidence:** 4

**Summary:**

This work summarizes the trend in DNN models of biological vision that networks that perform better on imagenet no longer necessarily provide better fits to neural data. It also shows neural-harmonized models do provide better fits to a data of mostly face-selective neurons.

**Strengths:**

Neural harmonizing makes a substantial impact on neural fit.

Understanding what makes DNNs stop fitting neural data is an important problem

The use of a more naturalistic color images is well-motivated

**Weaknesses:**

The framing of the paper (particularly the title) suggests that this paper is making a novel claim about DNN performance, when in fact this claim is based on a re-plot of BrainScore data and has been made before (in ref 11 and here: https://www.biorxiv.org/content/10.1101/688390v1). While it is good to have a replication of this finding, the substantial novel contribution of this paper is to show that neural harmonizing increases the match to neural data, so the title and framing should reflect that.

The measure of relevant features is too anecdotal as presented currently to support the claims (see below)

**Questions:**

line 58 has a typo

The abstract says "Our results suggest that harmonized DNNs break the trade-off between ImageNet accuracy
and neural prediction accuracy that assails current DNNs and offer a path to more accurate models of biological vision. " I'm not sure what tradeoff is being referred to here. Is it that higher accuracy leads to worse neural prediction? That tradeoff is not broken by using the harmonizer. The harmonized models have better predictivity but not better Imagenet performance.

Figure 4 shows feature relevance maps for example images, but without seeing this quantified across a large number of images, not much can be taken from this. The authors also make reference to models paying too much attention to background, etc, but again this is not measured in any objective way and seems to just be based on looking at these examples. Can the authors quantify these claims and show they hold across large populations of images?

In fig 5, why do many of the features have so little relative importance? Are these meant to be the top features? Also, why in the lower plot are there 4 colored bars but only three colored boxes? The axes are also different for these plots. Is there any significance to the fact that the relative importance for features from the harmonized model is half that of the resnet model? What else explains the responses and do those other features look like those from the unharmonized model? In total, I don't feel I can take any strong results away from this plot as-is.

**Limitations:**

The authors address limiations

---

> ### Author Rebuttal · Authors · 2023-08-09
>
> > The framing of the paper (particularly the title) suggests that this paper is making a novel claim about DNN performance, when in fact this claim is based on a re-plot of BrainScore data and has been made before (in ref 11 and here: https://www.biorxiv.org/content/10.1101/688390v1). While it is good to have a replication of this finding, the substantial novel contribution of this paper is to show that neural harmonizing increases the match to neural data, so the title and framing should reflect that.
>
> Thank you for the comment. We have addressed this in the main rebuttal. We are copying the response below for your convenience. Please let us know if you have any further questions, concerns, or points for clarification about this.
>
> The correlation between the accuracy of DNNs on object recognition tasks like ImageNet and their ability to predict IT responses has been used as evidence for long-standing theories in vision science, such as core object recognition [1] (our title is an homage to this paper). As we try to emphasize in our submission, this correlation has not only weakened in recent years [2] for nonhuman primate electrophysiology, it has begun to progressively **worsen** as DNNs improve on ImageNet — especially over the last four years as DNNs have begun to dramatically increase in scale. We will clarify that it is this worsening-with-accuracy neural alignment of DNNs that we believe is truly novel (and alarming!). We will adopt the reviewers’ suggestions for clarifying that the results scraped from the Brain-Score website are potentially an “observation by many” (even if it is not yet a published finding) vs. the data we introduce and model is a “finding by the authors”.
>
> The reviewers note that similar issues with performance optimization as we describe have been discussed in the human fMRI literature [3]. We will include a section on human fMRI in our revision as well as an elaboration on the important differences regarding inferences about neural computations based on electrophysiology (as we do) vs. fMRI (which at best offers a very slow and indirect readout of neural populations [4]).
>
> > I'm not sure what tradeoff is being referred to here... That tradeoff is not broken by using the harmonizer. The harmonized models have better predictivity but not better Imagenet performance.
>
> We apologize for the lack of clarity. As non-harmonized models have improved in ImageNet accuracy, they have become progressively worse at predicting neural responses. In contrast, we see a mostly significant linear trend between the ImageNet accuracy and neural prediction accuracy of harmonized DNNs (Monkey 1, PL: $\rho = 0.37$, $p < 0.01$, Monkey 2, PL: $\rho = 0.23$, $p < 0.05$, Monkey 1, ML: $\rho = 0.15$, $n.s.$). Moreover, nearly all harmonized DNNs outperform their nonharmonized baselines on ImageNet accuracy. We will clarify this in the manuscript.
>
> > Figure 4 shows feature relevance maps for example images...
>
> We apologize for the confusion. The results in Figure 3 of our submission depict the average correlation between spatially-resolved predictions of neural activities across images and the ground truth responses. In Figure 4 we plot maps of model predictions and true neural responses for several images. These are not saliency/gradient/feature attribution maps commonly used in explainable AI, but actual predictions and ground truth of the neural activity evoked by different regions of images. Our subjective interpretations of these spatial maps are that non-harmonized DNN responses are driven more by background features than harmonized models. We will clarify.
>
> > Questions about Figure 5.
>
> We used CRAFT [5] to identify the primary features relied on by harmonized/unharmonized ResNet50s to predict IT image-evoked responses. To do this, CRAFT first computes non-negative matrix factorization to find features, and then total Sobol indices to measure the relative importance of each feature. Total Sobol indices provide a score representing the proportion of variance attributed to each individual concept and its interactions. These indices do not always sum to 1 because of how feature interactions are accounted for by total Sobol indices. For example, the importance of Feature 1 includes an interaction between Feature 1 and Feature 2, which is also captured within Feature 2. This leads to redundancy, which explains why the total Sobol indices can range between 0 and 1 without always summing to 1. We will revise our explanation of this procedure for clarity.
>
> We also used CRAFT to find image patches that depict features of various importance. The bar plot colors in Fig. 4 denote relative importance features described by image patches on the left. (Note that we inadvertently highlighted 4 bars for unharmonized models; this should be 3.) CRAFT thus offers a principled and qualitative explanation of what features drive models' neural activity predictions.
>
> > line 58 has a typo
>
> We fixed this. Thanks!
>
>
> [1] Yamins, D.L.K., Hong, H., Cadieu, C.F., Solomon, E.A., Seibert, D., DiCarlo, J.J.: Performance-optimized hierarchical models predict neural responses in higher visual cortex. Proc. Natl. Acad. Sci. U. S. A. 111(23) 307 (June 2014) 8619–8624
>
> [2] Schrimpf, M., Kubilius, J., Hong, H., Majaj, N.J., Rajalingham, R., Issa, E.B., Kar, K., Bashivan, P., 327 Prescott-Roy, J., Geiger, F., Schmidt, K., Yamins, D.L.K., DiCarlo, J.J.: Brain-Score: Which artificial neural network for object recognition is most Brain-Like? (January 2020)
>
> [3] Jozwik, K., Schrimpf, M., Kanwisher, N., Dicarlo, James. 2019. To find better neural network models of human vision, find better neural network models of primate vision. BioRxiv.
>
> [4] Heeger, D., Ress, D. 2002. What does fMRI tell us about neuronal activity? Nature reviews neuroscience.
>
> [5] Fel, T., Picard, A., Bethune, L., Boissin, T., Vigouroux, D., Colin, J., Cadène, R., Serre, T. 2023. CRAFT:
> Concept recursive activation FacTorization for explainability. CVPR.

---

> > ### Comment · Reviewer_X5jQ · 2023-08-14
> >
> > I appreciate the authors' response and clarifications. They did not address my concern about the interpretation of Figure 3 being based on limited/subjective data. It should also be noted that ref [6] in the general response notes a negative correlation between performance and fit (i.e., the idea that better imagenet models are worse brain models is noted there). I will leave my score as-is.

---

> > > ### Author Response · Authors · 2023-08-14
> > > **Response**
> > >
> > > Thanks so much for the response. Sorry also about any confusion. Let us clarify:
> > >
> > > > They did not address my concern about the interpretation of Figure 3 being based on limited/subjective data.
> > > In the original response, the reviewer said, **Figure 4 shows feature relevance maps for example images, but without seeing this quantified across a large number of images, not much can be taken from this.** Are you referring to Figure 3 or 4 here?
> > >
> > > Figure 3 shows the average + error bars of the correlations between each models' predictions of neuronal activity for images and the actual neuronal responses to those images. As discussed in the text, there are highly statistically significant differences between models. In our response to X5JQ we also noted that the correlations of harmonized model neural predictivity vs. ImageNet accuracy are also significantly positive for 2/3 monkey/area combinations. Thus, the dataset is statistically powered enough for hypothesis testing. Figure 4 just shows what these predictions look like. Can you please clarify what is subjective here, and what is making you hesitate in improving your review?
> > >
> > > > It should also be noted that ref [6] in the general response notes a negative correlation between performance and fit...
> > >
> > > Ref [6] from your response indeed shows a negative correlation between ImageNet accuracy and "human IT predictivity." But as we mentioned in our response, these are results reflect human fMRI data not electrophysiology data (like our paper features). The authors' of that paper also use representational dissimilarity (RDM) to measure the similarity between models and fMRI activity, whereas we measure the correlation between model predicted and real image-evoked neuronal activity.
> > >
> > > There is no guarantee that results in fMRI will generalize to electrophysiology, and in fact they often do not (see ref [4] from our response). As mentioned in our response, we thought that adding an fMRI section into the discussion of our paper would be a good compromise, to show that while some have found that fMRI fits vs. ImageNet accuracy have decreased over the years (like in ref [6] from your response), others like [1] below have found that is not always the case when using fMRI and RSA. In contrast, in our manuscript we report that six different electrophysiological recordings of nonhuman primate show the same increase-then-decrease of ImageNet accuracy vs. neuronal prediction accuracy.
> > >
> > > [1] Conwell C., Prince J., Kay K., Alvarez G., Konkle T. 2023. What can 1.8 billion regressions tell us about the pressures shaping high-level visual representation in brains and machines? BioRxiv.
> > >
> > > Thank you again for engaging and we hope to continue this discussion!

---

> > > > ### Comment · Reviewer_X5jQ · 2023-08-14
> > > >
> > > > Sorry, yes Figure 4. The description of Figure 4 results (198-205) talks about certain models relying on background features versus not etc., but I can't tell if these are robust claims, or based simply on the example images given here.

---

> > > > > ### Author Response · Authors · 2023-08-14
> > > > > **Response**
> > > > >
> > > > > Got it, thanks. Yes these are interpretations of the predictions of each model in Figure 4, along with interpretations of the most-important features according to CRAFT in Figure 5. In any case, we agree that there's room for an additional layer of quantification, by segmenting scenes into foreground/background elements, and seeing how harmonized vs. unharmonized model neuronal predictions correlate with each. We can add this as a future direction that would significantly help Neuroscience researchers use harmonized models to generate predictions on the feature dimensions encoded in their neuronal recordings. Thanks again!

---

### Official Review · Reviewer_nyjr · 2023-07-07

**Soundness:** 3 good
**Presentation:** 3 good
**Contribution:** 3 good
**Rating:** 7
**Confidence:** 4

**Summary:**

This paper investigates the general finding that modern deep neural network architectures have worse predictions of primate IT cortex, even though they perform better at object recognition. The authors investigate a set of IT recordings that incorporate spatially resolved population maps, and show that the best fitting units of modern DNNs have different spatial activity than the neuron they are predicting. The authors investigate a set of models that are trained with a “neural harmonizer” that helps align the model responses with human behavior and find that this training significantly improves the neural predictivity for many different architectures and also results in more similar spatial activation maps.

Edit after rebuttal period: I read the author’s rebuttal and resulting discussion about my concerns. Specifically, it was good to see the preliminary results showing that the harmonized models do not impair the predictivity for the public BrainScore IT neural dataset. These results better contextualize the claims by the authors. I would hope that the full analysis can be completed on the rest of the datasets but even just adding this preliminary data contextualizes their claims, and given this I updated my score.

**Strengths:**

* The authors analyze neural predictivity with a dataset that has not been extensively studied before, and analyze a large number of candidate neural networks varying in architecture and training procedure.
* The authors investigate models trained on behavioral data in addition to ImageNet. Although these models were presented last year at NeurIPS, to my knowledge, their evaluations on neural data had not been reported, and the increase in predictivity with the behavioral data as a regularizer is impressive. I think makes the contribution novel enough even if the neural harmonizer methodology was presented before, as very few models have explicitly attempted to match precise human behavioral results as a way of increasing neural alignment.
* Even though it is generally known that with modern models improved ImageNet performance does not lead to better brain predictions (further discussed in Weaknesses below), the full investigation and quantification of this in this paper is potentially beneficial to the field as a way to formally state that this trend is no longer the case.
* The analysis of spatially-mapped neural responses is novel, and the breakdown of which parts of the image are most important for explaining the neural data is interesting.

**Weaknesses:**

1) The authors present the lack of a correlation between neural predictivity and brain responses for modern models as a new finding, however this is a generally known phenomena mentioned in various papers as motivation for developing better metrics of similarity. Additionally, one can see this is the case by simply looking at the brain-score website. The authors include a footnote that in [11] this was mentioned, but it might be more appropriate to reframe the beginning of the abstract and the introduction so that this is less of a “finding by the authors” and more of an “observation by many in the field”.

2) The authors state that “task optimization” is insufficient for reverse-engineering IT, however the only “task” that they consider is ImageNet. There are many other “tasks” that could be considered to train neural network models that are different from ImageNet (for instance including auxiliary tasks) and may lead to better predictions, so this seems like too strong of a claim.

3) Similarly the claim that DNNs *need* biologically-aligned training routines seems too strong. This is one way of getting to improved predictivity, but it may not be the only way (and in fact, the predictions are still well below the noise ceiling).

**Questions:**

a) Can results from the brain-score benchmarks (Figure 1) be included for the neural harmonizer models that are showed in Figure 3? These datapoints would further support the claims, especially given that the neural data analyzed is primarily from face-patches. If the trend does not hold in those datasets how does this change the conclusions?

b) Is cross validation performed by holding out an entire image or by holding out patches? This potentially matters for interpretation, as some of the image statistics may be very similar for neighboring patches of the image.

c) I’m a little confused by the wording in lines 54-57. Is (i) about the training data and (ii) about the architecture? If so, could this be made explicit?

d) In line 149 it is stated that separate fitting procedures were performed for every layer of activities and the best layer fit is reported – was this best layer determined using a separate “validation” set of data and the reported neural predictivity score is from “test” data? If not, there are some “double dipping” concerns for this type of analysis.

**Limitations:**

The authors include sections on limitations and broader impacts.

---

> ### Author Rebuttal · Authors · 2023-08-09
>
> > The authors present the lack of a correlation between neural predictivity and brain responses for modern models as a new finding, however this is a generally known phenomena mentioned in various papers as motivation for developing better metrics of similarity.
>
> Thank you for this comment. We addressed this point in our main rebuttal, but to quickly reiterate: we have taken your comment to heart and will be more precise about the novelty of our contribution, and clarify which findings were scraped and potentially an observation by many vs. truly novel results from our dataset and modeling efforts.
>
> > The authors state that “task optimization” is insufficient for reverse-engineering IT, however the only “task” that they consider is ImageNet.
>
> This is a very important point, thank you! As discussed in the rebuttal, we expanded our analyses to include DNNs trained on other tasks and datasets (Taskonomy and Ecoset). None of these models are as accurate as ImageNet-trained or Harmonized models in predicting image-evoked responses in our recordings. With that said, we agree that there are many other potential tasks and datasets out there, and we will soften our claims as a result. In our revision, we will state that our harmonized DNNs are more accurate in predicting neural responses than any of the 135 ImageNet-trained, 19 Taskonomy-trained, or 4 Ecoset-trained DNNs that we tested.
>
> > Similarly the claim that DNNs need biologically-aligned training routines seems too strong.
>
> We completely agree that biologically-aligned training routines may not be sufficient to reliably predict neural responses. However, our findings indicate that alignment with human behavior is the current best approach to achieving this goal. We will soften our claims to make it clear that harmonization and biologically-aligned training routines are not the only way (and maybe not the best way) to build better models of neural function.
>
> > Was [the] best layer determined using a separate “validation” set of data and the reported neural predictivity score is from “test” data?
>
> Thank you for this question, which we addressed in our main rebuttal, and copy below. Please let us know if you have any further questions or concerns.
>
> Our neural data fitting procedure followed the original Brain-Score [1] procedure precisely, “The final neural predictivity score for the target brain region is computed as the mean across all train-test splits.” In other words, we (and the standard Brain-Score) did not use a held-out validation set to select the most predictive layer for each model. We agree with the reviewer, however, that this approach is potentially flawed. To test for this possibility we redid our analyses with train/val/test dataset partitions and selected a layer for each model as whichever one achieved the best performance on the validation set. For this procedure, we once again held one image out for testing, but in this version of the analysis, we also held out half of the training data for layer selection (validation). Our results from this approach strongly correlated with the ones reported in our original submission, and our overall conclusions remain the same (Monkey 1 ML Rebuttal Fig. A, $\rho = 0.98$, $p < 0.001$, Monkey 2 ML Rebuttal Fig. B, $\rho = 0.97$, $p < 0.001$, Monkey 1 PL, $\rho = 0.98$, $p < 0.001$). In our revision, we will report results from these complete cross-validation analyses instead of the standard Brain-Score analyses.
>
> > Is cross validation performed by holding out an entire image or by holding out patches?
>
> We always tested on held-out images. In our cross-validation procedure described above, our validation set consisted of held-out patches. As this is just for validation, we do not believe it confounds interpretation.
>
> > I’m a little confused by the wording in lines 54-57.
>
> In those lines, (i) is about training data, objective functions, optimizers, learning algorithms (e.g., backpropagation or not?), and other choices that shape training that are unrelated to model architecture. (ii) Is strictly about model architecture, which can be induced with brain-like constraints, like normalizations and recurrence. We will rewrite these lines for clarity.
>
> > Can results from the brain-score benchmarks (Figure 1) be included for the neural harmonizer models that are shown in Figure 3? These data points would further support the claims, especially given that the neural data analyzed is primarily from face-patches. If the trend does not hold in those datasets how does this change the conclusions?
>
> As discussed in the Contributions section of our submission, we focused on IT recordings from [1] because, “[These] experimental images were significantly closer to the statistical distribution of images in ImageNet (Fig. S1.), unlike [IT recordings used in Brain-Score][2].” The publicly available IT data on Brain-Score.org is of responses to images that are out-of-distribution of ImageNet (Fig. 1a and Fig. S1; Fig. 1b and Fig. 1c feature scraped results of DNN scores on private image datasets). Because of the well-known sensitivity of DNNs to distributional shifts [3], we believe it will be difficult to interpret the results of harmonized DNNs on these data. For this reason, we prefer to focus on our dataset from [1] in this manuscript. With that said, we included a link to our codebase on line 276 to make our data and models available to the wider community for additional analyses.
>
> [1] Arcaro, M.J., Ponce, C., Livingstone, M.: The neurons that mistook a hat for a face. Elife (June 2020)
>
> [2] Majaj, N.J., Hong, H., Solomon, E.A., DiCarlo, J.J.: Simple learned weighted sums of inferior temporal neuronal firing rates accurately predict human core object recognition performance. J. Neurosci. 35(39) (September 2015) 13402–13418
>
> [3] Geirhos, R., Medina Temme, C.R., Rauber, J., Schütt, H.H., Bethge, M., Wichmann, F.A.: Generalisation in humans and deep neural networks.

---

> > ### Comment · Reviewer_nyjr · 2023-08-14
> > **response to author rebuttal**
> >
> > Thank you for addressing my (and other reviewers) concerns and for making improvements to the paper based on these comments. The update with the choice of layers and the clarification about holding out images solidifies my confidence that the presented results make sense.
> >
> > However, I don't follow the logic behind why distribution shifts would impact the interpretation of the neural harmonizer models on other datasets. If it is the case that the neural harmonization is dataset specific, then this is a major limitation of the method and should be addressed. For instance, if the result on the BrainScore datasets looks different than the result on the presented dataset, that in itself is interesting and, in my opinion, should be presented for completeness. After all, a good model of the brain should match responses to ALL images, and not just subsets.  This seems like something that a lot of readers will be asking given that the paper starts out with an analysis from the BrainScore website (and one could submit the models to BrainScore to get the values for the private data). Primarily due to this, I am maintaining my original score.

---

> > > ### Author Response · Authors · 2023-08-14
> > > **Response**
> > >
> > > We totally agree that measuring out-of-distribution predictions is an interesting problem! We see it as a potentially critical difference between DNN models of visual perception and neural responses and actual biological vision. But we also believe that the first step towards rigorously testing out-of-distribution performance is having a foundation for within-distribution performance. Our datasets fill this surprising gap in the field, as the publicly available IT data on the Brain-Score website are out-of-distribution for ImageNet (Appendix Fig. 1 of our submission).
> > >
> > > As a next step and for future work, we believe one or multiple papers can be devoted to analyzing within vs. out-of-distribution affects on neuronal response prediction. Doing this analysis rigorously requires systematic control over how out-of-distribution a neural dataset is vs. ImageNet, and relating that difference to prediction accuracy. Achieving this goal is not possible without gathering new data, but we agree that it is a fascinating direction and will add it to our Discussion as future work. Thanks!!

---

> > > > ### Comment · Reviewer_nyjr · 2023-08-19
> > > >
> > > > I think I'm not understanding something correctly here. I agree that the question of in-distribution vs. out-of-distribution is interesting as a separate paper, but I don't understand why that would prevent showing the data for the Brain-Score datasets here for completeness? Could you elaborate?
> > > >
> > > > For instance, all of the models in Figure 1 of the paper are tested on the Brain-Score dataset even though most were trained on ImageNet, and naively I would expect them to have the same biases due to being tested on out-of-distribution data (so it seems fair to compare the neural harmonizer models to these other presented models). Additionally, as mentioned in the limitations, faces are also out of distribution from ImageNet (although using different measures of OOD than that presented in App. Fig 1), so in a sense this is also a problem for the datasets used in the paper.
> > > >
> > > > If it is the case that "neural harmonization" (which I believe is just based on behavioral responses, so doesn't require explicit training for the neural dataset) only improves predictions on some neural datasets but not others, then that seems like an important thing to include in this paper. Future work can dig into the "why" behind this, but it is a clear limitation that should be included so that the results are not over-extrapolated to all datasets.

---

> > > > > ### Author Response · Authors · 2023-08-19
> > > > > **Response**
> > > > >
> > > > > Thanks for pushing on this. We agree it's an important question.
> > > > >
> > > > > **Additionally, as mentioned in the limitations, faces are also out of distribution from ImageNet (although using different measures of OOD than that presented in App. Fig 1), so in a sense this is also a problem for the datasets used in the paper.**
> > > > >
> > > > > As mentioned in our discussion, faces are not a category in ImageNet, but ImageNet is replete with human and animal faces, and thus the feature is not OOD. Moreover, the experimental data we use shows that putative face patches in monkeys are highly responsive to non-facial features. So to the extent that our data is OOD, it is not clear that this could systematically increase the gap between harmonized/unharmonized models.
> > > > >
> > > > > **I don't understand why that would prevent showing the data for the Brain-Score datasets here for completeness... If it is the case that "neural harmonization" (which is just based on behavioral responses) only improves predictions on some neural datasets but not others, then that seems like an important thing to include in this paper. Future work can dig into the "why" behind this, but it is a clear limitation that should be included so that the results are not over-extrapolated to all datasets.**
> > > > >
> > > > > We can absolutely include results on the Majaj et al. 2015 data from Brain-Score in our Appendix. To clarify, our hesitation for including these is that it is conceptually difficult to directly compare those results to the recordings we analyze in our paper. We know there's a distribution shift between Majaj et al. 2015 and ImageNet, but we don't know *why* that shift occurs; hence we cannot easily explain the reason for any improvement or decrement in performance for harmonized models. That said, we see your perspective as well, and we will work on adding these results ASAP.

---

> > > > > > ### Comment · Reviewer_nyjr · 2023-08-19
> > > > > >
> > > > > > Thank you for the response and for attempting to get these results. If the results show that neural harmonization does not improve predictions on these datasets, then sections of the paper stating that neural harmonization breaks the trade off between imagenet accuracy and neural predictions should be updated.
> > > > > >
> > > > > > And I don't mean to seem nit-picky here, but why only the Majaj et al. 2015 data? If the models are submitted to the Brain-Score platform for evaluation (the way to evaluate on the private datasets, which should be used for comparison to the other models in Fig 1) then to my knowledge all of the datasets should be available?

---

> > > > > > > ### Author Response · Authors · 2023-08-19
> > > > > > > **Response**
> > > > > > >
> > > > > > > **If the results show that neural harmonization does not improve predictions on these datasets, then sections of the paper stating that neural harmonization breaks the trade off between imagenet accuracy and neural predictions should be updated.**
> > > > > > >
> > > > > > > We want to be on the same page about the purpose of these results. Each of the Brain-Score datasets is OOD in features vs ImageNet. The purpose of our paper is to understand why more ImageNet accurate DNNs are starting to perform worse in neural prediction accuracy. One obvious explanation is that the datasets used for testing on Brain-Score are OOD. The recordings we investigate do not have this same problem at a feature level, so our conclusion is straightforward: human behavior provides an important constraint on DNN training on ImageNet (via the neural harmonizer) that aligns their features with brains.
> > > > > > >
> > > > > > > For this reason we will mention these Brain-Score experiments in the Discussion as an empirical basis for future directions -- as you suggested -- and provide analyses and additional commentary in the Appendix. But to be clear these Brain-Score experiments are additions that our main conclusions do not depend on. Please advise.
> > > > > > >
> > > > > > > **Why only Majaj?**
> > > > > > > We can perform evaluations through the Brain-Score website to get harmonized/unharmonized model scores on the three IT datasets they have. Stay tuned, we are working on this.

---

> > > > > > > > ### Comment · Reviewer_nyjr · 2023-08-19
> > > > > > > >
> > > > > > > > >Each of the Brain-Score datasets is OOD in features vs ImageNet. The purpose of our paper is to understand why more ImageNet accurate DNNs are starting to perform worse in neural prediction accuracy.
> > > > > > > >
> > > > > > > > Yes, I understand that. But what I do not understand is why the neural harmonizer models should be treated differently than the standard models for evaluation when predicting neural data. Naively, it seems like ALL models trained on ImageNet will have this same problem (which I have stated other times in this thread). Because of this, I think that the main conclusions DO depend on the results.

---

> > > > > > > > > ### Author Response · Authors · 2023-08-19
> > > > > > > > > **Response**
> > > > > > > > >
> > > > > > > > > We really appreciate the continued discussion and opportunities to clarify!
> > > > > > > > >
> > > > > > > > > **But what I do not understand is why the neural harmonizer models should be treated differently than the standard models for evaluation when predicting neural data.**
> > > > > > > > > Hopefully we are not misunderstanding you here, but to be clear, we treat the harmonized and unharmonized DNNs exactly the same in all of our experiments.
> > > > > > > > >
> > > > > > > > > **Because of this, I think that the main conclusions DO depend on the results [of harmonized DNNs on the OOD Brain-Score datasets].**
> > > > > > > > > - We investigate dwhy it looks like DNNs are becoming progressively different from IT as they improve on ImageNet. (**Line 15**: *To understand why DNNs experience this trade-off and evaluate if they are still an appropriate paradigm for modeling the visual system, we turn to recordings of IT that capture spatially resolved maps of neuronal activity elicited by natural images [6].*)
> > > > > > > > >
> > > > > > > > > - We analyzed the specific neural recordings that we did (instead of the ones in Brain-Score) because ours measure spatial feature-level responses to images that are in the same distribution as ImageNet (**Line 63**: *The recordings also provided a coarse estimate of which image features drove neuronal activity (Fig. 2), which helped characterize DNN errors in explaining neural responses. As we will show, compared to the recordings used in the official Brain-Score benchmark, such spatially-resolved neural data provide far greater insight into why DNNs are becoming worse models of IT.*)
> > > > > > > > >
> > > > > > > > > - We then found on recordings of two monkeys in two separate regions of IT that the ImageNet/Neural prediction accuracy trade-off can be corrected by harmonization, which we show biases DNNs to rely on features that are more similar to IT. (**Line 76**: *We successfully broke this trade-off [in our recordings] by training DNNs with the neural harmonizer [7] and aligning the representations they learn for object recognition with those that are diagnostic for humans.*)
> > > > > > > > >
> > > > > > > > > - Throughout our manuscript we are careful to contrast our experiments with the datasets used in Brain-Score, and do not insist that Harmonization is the only way forward; only that our experiments suggest that it may be a partial solution to the ImageNet accuracy/neural prediction trade-off of DNNs (**Line 271**: *The neural harmonizer is a partial solution to the problems that DNNs face in modeling primate IT.*)
> > > > > > > > >
> > > > > > > > > Can you please clarify which of these conclusions are changed by these suggested harmonizer experiments on the Brain-Score datasets? We are on board with adding these experiments and results to the Appendix; we totally agree with your logic in that these would be exciting for future work! But as mentioned earlier in the thread (and as is our motivation for the paper), they are difficult to interpret, and so we are struggling to see how they could change our ultimate conclusions. Thanks!

---

> > > > > > > > > > ### Comment · Reviewer_nyjr · 2023-08-19
> > > > > > > > > >
> > > > > > > > > > >Hopefully we are not misunderstanding you here, but to be clear, we treat the harmonized and unharmonized DNNs exactly the same in all of our experiments.
> > > > > > > > > >
> > > > > > > > > > There is a bit of misunderstanding still. The following response was given to my question (a).
> > > > > > > > > >
> > > > > > > > > > >Because of the well-known sensitivity of DNNs to distributional shifts [3], we believe it will be difficult to interpret the results of harmonized DNNs on these data. For this reason, we prefer to focus on our dataset from [1] in this manuscript.
> > > > > > > > > >
> > > > > > > > > > Once again, **why should the harmonized DNNs be interpreted differently than the standard DNNs on other neural datasets**? ALL of the models will be sensitive to distribution shifts, and so any interpretation that one would make on the initial models (ie in Figure 1 of ImageNet accuracy vs. predictions) should also be applicable to the harmonized DNNs. If I am missing something here please explain.
> > > > > > > > > >
> > > > > > > > > > Regarding specific conclusions this would influence, it is this section:
> > > > > > > > > >
> > > > > > > > > > >We then found on recordings of two monkeys in two separate regions of IT that the ImageNet/Neural prediction accuracy trade-off can be corrected by harmonization, which we show biases DNNs to rely on features that are more similar to IT. (Line 76: We successfully broke this trade-off [in our recordings] by training DNNs with the neural harmonizer [7] and aligning the representations they learn for object recognition with those that are diagnostic for humans.)
> > > > > > > > > >
> > > > > > > > > > In the quoted text from Line 76 you seem to already be attempting to correct for this with the [in our recordings] addition. These caveats are incredibly important so that readers do not over generalize the results! If it is the case that the harmonized models are only better on this dataset but not others, then it needs to be stated clearly in the paper. One place that is similarly overstated are lines 21-24 of the abstract, which could be changed to:
> > > > > > > > > > "Our results suggest that [in some datasets] harmonized DNNs break the trade-off between ImageNet and ... "
> > > > > > > > > >
> > > > > > > > > > Note, this is assuming that the harmonization does not help with the other datasets! When I wrote my initial review I just assumed that it would help for those datasets too and the results would just strengthen the paper, but given the hesitation about reporting them I'm guessing that it doesn't?

---

> > > > > > > > > > > ### Author Response · Authors · 2023-08-19
> > > > > > > > > > > **Response**
> > > > > > > > > > >
> > > > > > > > > > > **In the quoted text from Line 76 you seem to already be attempting to correct for this with the [in our recordings] addition.**
> > > > > > > > > > > We added "[in our recordings]" to the quote of Line 76 because we believe that this intent is clear in context (Line 69 below; these are quotes from the contributions section). **Line 69**: *We observed the same trade-off we found on Brain-Score.org data (Fig. 1) in each of our recordings: DNNs are becoming less accurate at predicting IT responses as they improve on ImageNet (Fig. 3).*
> > > > > > > > > > >
> > > > > > > > > > > **Note, this is assuming that the harmonization does not help with the other datasets! When I wrote my initial review I just assumed that it would help for those datasets too and the results would just strengthen the paper, but given the hesitation about reporting them I'm guessing that it doesn't?**
> > > > > > > > > > > We genuinely have not run these experiments yet because we see them as difficult to interpret for the reasons we have discussed. We will follow up with these results as soon as we have them and include them in the Appendix as well as mention them in the Discussion.
> > > > > > > > > > >
> > > > > > > > > > > **Once again, why should the harmonized DNNs be interpreted differently than the standard DNNs on other neural datasets? ALL of the models will be sensitive to distribution shifts, and so any interpretation that one would make on the initial models (ie in Figure 1 of ImageNet accuracy vs. predictions) should also be applicable to the harmonized DNNs. If I am missing something here please explain.**
> > > > > > > > > > > We are treating *experimental datasets* differently, not models. Our conclusions are based on results from a zoo of 135 different DNNs tested on our spatially-resolved neural data (note that this is more than the number of models that were in the Brain-Score results we scraped). We have explained why the Brain-Score stimuli are challenging from an interpretation perspective for both DNN and neuronal responses, and thus are better used to motivate computational problems than as a resource for rigorously understanding neural coding. Indeed, we believe that our controlled experiments and interpretations are a feature of our work for a Neuroscience audience.

---

> > > > > > > > > > > > ### Comment · Reviewer_nyjr · 2023-08-19
> > > > > > > > > > > >
> > > > > > > > > > > > >In the quoted text from Line 76 you seem to already be attempting to correct for this with the [in our recordings] addition. We added "[in our recordings]" to the quote of Line 76 because we believe that this intent is clear in context (Line 69 below; these are quotes from the contributions section). Line 69: We observed the same trade-off we found on Brain-Score.org data (Fig. 1) in each of our recordings: DNNs are becoming less accurate at predicting IT responses as they improve on ImageNet (Fig. 3).
> > > > > > > > > > > >
> > > > > > > > > > > > I did not think that this was clear in line 76 or the noted lines 21-24 from the abstract. Due to this, when I read the paper I expected to see an updated version of Figure 1 with the harmonized models.
> > > > > > > > > > > >
> > > > > > > > > > > > >We are treating experimental datasets differently, not models. Our conclusions are based on results from a zoo of 135 different DNNs tested on our spatially-resolved neural data (note that this is more than the number of models that were in the Brain-Score results we scraped). We have explained why the Brain-Score stimuli are challenging from an interpretation perspective for both DNN and neuronal responses, and thus are better used to motivate computational problems than as a resource for rigorously understanding neural coding. Indeed, we believe that our controlled experiments and interpretations are a feature of our work for a Neuroscience audience.
> > > > > > > > > > > >
> > > > > > > > > > > > Just to clarify, I am not trying to suggest that the dataset used in this paper is worse than that used on Brain-Score (in fact, if you look at the listed strengths I very much enjoyed this paper because of the experimental data which has not been used for this type of analysis before, and that is still pushing it over the publication threshold for me). That said, I think it is very important as scientists to relate our findings back to work that has previously been done. The very first figure in your paper starts with the Brain-Score data. My impression of the responses in this thread was that you were saying the Brain-Score datasets were harder to interpret *for the harmonized DNN models compared to the standard models*, due to the following line:
> > > > > > > > > > > >
> > > > > > > > > > > > >Because of the well-known sensitivity of DNNs to distributional shifts [3], we believe it will be difficult to interpret the results of harmonized DNNs on these data. For this reason, we prefer to focus on our dataset from [1] in this manuscript.
> > > > > > > > > > > >
> > > > > > > > > > > > Maybe this is a misunderstanding and you are trying to state that ALL results from Brain-Score datasets are hard to interpret? I don't disagree, but given the prevalence of the datasets in previous work (and the fact that you are already using them in Figure 1 for motivation) it just seemed worth closing the circle and testing your additional models on the same data because otherwise readers (like myself) will naturally wonder why it wasn't presented.
> > > > > > > > > > > >
> > > > > > > > > > > > All of this being said, I also understand that running these additional evaluations is time consuming and may very well not be possible during the remaining response period. I hope that you consider including these results in the final submission as I think it will further complete the story that many readers will be asking.

---

> > > > > > > > > > > > > ### Comment · Reviewer_itup · 2023-08-20
> > > > > > > > > > > > >
> > > > > > > > > > > > > I've been following this conversation, and I agree with **nyjr** it is worth at least acknowledging the potential for this method to only work on certain datasets due to image distribution differences or differences in regions they target.
> > > > > > > > > > > > >
> > > > > > > > > > > > >
> > > > > > > > > > > > > I believe the authors should also consider citing the following paper which is similar in approach to their paper, albeit investigating a totally different issue:
> > > > > > > > > > > > >
> > > > > > > > > > > > > - Do Perceptually Aligned Gradients Imply Adversarial Robustness? (Roy Ganz, Bahjat Kawar, Michael Elad)

---

> > > > > > > > > > > > > > ### Author Response · Authors · 2023-08-20
> > > > > > > > > > > > > > **Response to itup**
> > > > > > > > > > > > > >
> > > > > > > > > > > > > > **I've been following this conversation, and I agree with nyjr it is worth at least acknowledging the potential for this method to only work on certain datasets due to image distribution differences or differences in regions they target.**
> > > > > > > > > > > > > > Thank you for this comment! We agree that we can be more precise about the scope of our work as it currently stands, and we will revise our paper to do this. We are also running the Brain-Score experiments that **nyjr** requested, and will take those into account with this revision.
> > > > > > > > > > > > > >
> > > > > > > > > > > > > > **I believe the authors should also consider citing the following paper which is similar in approach to their paper, albeit investigating a totally different issue: Do Perceptually Aligned Gradients Imply Adversarial Robustness? (Roy Ganz, Bahjat Kawar, Michael Elad)**
> > > > > > > > > > > > > > Thank you for this reference. We will incorporate it in our revision.

---

> > > > > > > > > > > > > > > ### Author Response · Authors · 2023-08-22
> > > > > > > > > > > > > > > **Response to nyjr and itup**
> > > > > > > > > > > > > > >
> > > > > > > > > > > > > > > Thanks again for the great discussion.
> > > > > > > > > > > > > > >
> > > > > > > > > > > > > > > We have been working hard to evaluate our model zoo with the Brain-Score API. Unfortunately, there is a backend error that is stopping us from uploading our models to their server for evaluation. We are in touch with the Brain-Score organizers to fix this problem and will update you both with results as soon as we can.
> > > > > > > > > > > > > > >
> > > > > > > > > > > > > > > In the meantime, we evaluated harmonized and unharmonized models on the validation split of the publicly available IT dataset (Majaj et al., 2015). We report these results in the table below:
> > > > > > > > > > > > > > >
> > > > > > > > > > > > > > > | Model | **Neural Harmonized** | **Neural Baseline** | **Neural difference** | **ImageNet Harmonized** | **ImageNet Baseline** | **Accuracy difference** |
> > > > > > > > > > > > > > > |---|---|---|---|---|---|---|
> > > > > > > > > > > > > > > | **VGG** | 0.52356419 | 0.52257217 | 0.52356419 | 69.369 | 67.23 | 2.139 |
> > > > > > > > > > > > > > > | **ResNet50v2** | 0.50731932 | 0.5009733 | 0.00634602 | 77.17 | 76 | 1.17 |
> > > > > > > > > > > > > > > | **ConvNext** | 0.52366153 | 0.51967374 | 0.00398779 | 75.8 | 74.3 | 1.5 |
> > > > > > > > > > > > > > > | **MaxViT** | 0.54332366 | 0.5408319 | 0.00249176 | 78.7 | 78.6 | 0.1 |
> > > > > > > > > > > > > > > | **EfficientNetB0** | 0.50847317 | 0.50690659 | 0.00156658 | 77.51 | 75.4 | 2.11 |
> > > > > > > > > > > > > > > | **LeViT** | 0.4734579 | 0.4725359 | 0.000922 | 75.4 | 74.9 | 0.5 |
> > > > > > > > > > > > > > > | **ViT** | 0.4995311 | 0.4936421 | 0.005889 | 75.7 | 73.2 | 2.5 |
> > > > > > > > > > > > > > >
> > > > > > > > > > > > > > > For each model, Harmonization improves DNN accuracy in predicting neural responses on this dataset (albeit by a very small margin). It is difficult to explain why the size of the margin of improvement for harmonized models has decreased due to the stimuli and experimental design of this dataset (as we have discussed). We will include a discussion of these results as well as an expanded analysis on all Brain-Score benchmarks once they fix their backend bugs in our revision.
> > > > > > > > > > > > > > >
> > > > > > > > > > > > > > > The main conclusions of our paper have not changed with these results. Aligning DNNs with human behavior improves their ability to predict primate image evoked responses in primate IT. For our experiments, such alignment even corrects the trade-off between ImageNet and neural prediction accuracy that unharmonized DNNs experience. However, these results on the Brain-Score recordings make it clear that such benefits from Harmonization are by no means guaranteed and they appear to fall off for stimuli that are out-of-distribution. Future work will be needed to build more effective approaches for alignment than Harmonization, and more reliable and interpretable experiments for understanding the effects of distribution shifts on IT neuronal responses.
> > > > > > > > > > > > > > >
> > > > > > > > > > > > > > > We will update our manuscript to reflect these results and conclusions.

---

> > > > > > > > > > > > > ### Author Response · Authors · 2023-08-20
> > > > > > > > > > > > > **Response to nyjr**
> > > > > > > > > > > > >
> > > > > > > > > > > > > **Maybe this is a misunderstanding and you are trying to state that ALL results from Brain-Score datasets are hard to interpret? I don't disagree, but given the prevalence of the datasets in previous work (and the fact that you are already using them in Figure 1 for motivation) it just seemed worth closing the circle and testing your additional models on the same data because otherwise readers (like myself) will naturally wonder why it wasn't presented.**
> > > > > > > > > > > > > Thank you for all of your adroit comments, and especially this one! We do believe that because the stimuli used in the Brain-Score IT recordings are OOD of ImageNet, they can present interpretation challenges for DNNs trained on ImageNet. This is especially true for our work, where we are trying to understand why DNNs are becoming less accurate at predicting neural responses as their accuracy on ImageNet increases. Are the latest and greatest DNNs now learning features that are misaligned with IT? Or is there some other, potentially low-level explanation, that can be attributed to the distribution shift? We believe your arguments are very well reasoned in all of this and agree that adding the additional Brain-Score analyses will add to the paper.
> > > > > > > > > > > > >
> > > > > > > > > > > > > **All of this being said, I also understand that running these additional evaluations is time consuming and may very well not be possible during the remaining response period. I hope that you consider including these results in the final submission as I think it will further complete the story that many readers will be asking.**
> > > > > > > > > > > > > We appreciate this! We are trying to finish this before the discussion period ends. Even if we can't finish by then, we will incorporate these results in the final submission.
> > > > > > > > > > > > >
> > > > > > > > > > > > > **I did not think that this was clear in line 76 or the noted lines 21-24 from the abstract. Due to this, when I read the paper I expected to see an updated version of Figure 1 with the harmonized models.**
> > > > > > > > > > > > > We apologize that we were not clearer. We will revise the wording and adjust the scope depending on the outcomes of these Brain-Score experiments.

---

### Author Rebuttal · Authors · 2023-08-09

# Response to all reviewers

We thank the reviewers for their extensive feedback. We are confident that we have addressed their main critiques, which we summarize below along with our responses (the relevant reviewers are in parentheses):

**(nyjr) How was the best layer for each model selected?**
> Our neural data fitting procedure followed the original Brain-Score [1] procedure precisely, “The final neural predictivity score for the target brain region is computed as the mean across all train-test splits.” In other words, we (and the standard Brain-Score) did not use a held-out validation set to select the most predictive layer for each model. We agree with the reviewer, however, that this approach is potentially flawed. To test for this possibility we redid our analyses with train/val/test dataset partitions and selected a layer for each model as whichever one achieved the best performance on the validation set. For this procedure, we once again held one image out for testing, but in this version of the analysis, we also held out half of the training data for layer selection (validation). Our results from this approach strongly correlated with the ones reported in our original submission, and our overall conclusions remain the same (Monkey 1 ML Rebuttal Fig. A, $\rho = 0.98$, $p < 0.001$, Monkey 2 ML Rebuttal Fig. B, $\rho = 0.97$, $p < 0.001$, Monkey 1 PL, $\rho = 0.98$, $p < 0.001$). We will report both versions of results in our revision.

**(nyjr, itup) The only task considered is ImageNet. Would other datasets and tasks yield different or better predictions of neural responses?**
> We have added results on the ability of DNNs trained on the Taskonomy dataset and task set [2] and also the Ecoset [3] naturalistic object categorization dataset and task. We took 19 DNNs pretrained on each task in the Taskonomy, and 4 DNNs trained for classification on ecoset, and applied the standard Brain-Score fitting procedure that we describe in our original submission Methods to derive prediction accuracy scores for each model. These models were far less effective at explaining neural activity than any of the ImageNet-trained models we provided in the main text (Rebuttal tables C and D). **To summarize, our human behavior-aligned “Harmonized” DNNs are more accurate at predicting neural responses than any other DNN dataset or task we tested.**

**(nyjr, X5jQ) How novel is the top-line finding that ImageNet optimization is no longer effective for systems identification of primate IT?**
> The correlation between the accuracy of DNNs on object recognition tasks like ImageNet and their ability to predict IT responses has been used as evidence for long-standing theories in vision science, such as core object recognition [4] (our title is an homage to this paper). As we try to emphasize in our submission, this correlation has not only weakened in recent years [5] for nonhuman primate electrophysiology, it has begun to progressively **worsen** as DNNs improve on ImageNet. We will clarify that it is this worsening-with-accuracy neural alignment of DNNs that we believe is truly novel (and alarming!). We will adopt the reviewers’ suggestions for clarifying that the results scraped from the Brain-Score website are potentially an “observation by many” (even if it is not yet a published finding) vs. the data we introduce and model, which is a “finding by the authors”.

> The reviewers note that similar issues with performance optimization as we describe have been appeared in the human fMRI literature [6, 7], although those papers describe older network architectures from before 2022. We will include a section on human fMRI in our revision as well as an elaboration on the important differences regarding inferences about neural computations based on electrophysiology (as we do) vs. fMRI (which at best offers a very slow and indirect readout of neural populations [8]).

We have provided detailed comments to each of the reviewers’ responses below. We hope you agree that our manuscript has improved through your feedback and that our findings will have a significant impact on Computational Neuroscience.

[1] Schrimpf, M., Kubilius, J., Hong, H., Majaj, N.J., Rajalingham, R., Issa, E.B., Kar, K., Bashivan, P., 327 Prescott-Roy, J., Geiger, F., Schmidt, K., Yamins, D.L.K., DiCarlo, J.J.: Brain-Score: Which artificial 328 neural network for object recognition is most Brain-Like? (January 2020)

[2] Zamir, A., Sax, A., Shen, W., Guibas, L., Malik, J., Savarese, S. 2018. Taskonomy: Disentangling task transfer learning. IEEE Computer vision and pattern recognition conference (CVPR).

[3] Mehrer, J., Spoerer, C., Jobes, E., Kietzmann, T. 2021. An ecologically motivated image dataset for deep learning yields better models of human vision. Proc. Natl. Acad. Sci. U. S. A. 118(8).

[4] Yamins, D.L.K., Hong, H., Cadieu, C.F., Solomon, E.A., Seibert, D., DiCarlo, J.J.: Performance-optimized hierarchical models predict neural responses in higher visual cortex. Proc. Natl. Acad. Sci. U. S. A. 111(23) 307 (June 2014) 8619–8624

[5] Schrimpf, M., Kubilius, J., Hong, H., Majaj, N.J., Rajalingham, R., Issa, E.B., Kar, K., Bashivan, P., 327 Prescott-Roy, J., Geiger, F., Schmidt, K., Yamins, D.L.K., DiCarlo, J.J.: Brain-Score: Which artificial neural network for object recognition is most Brain-Like? (January 2020)

[6] Jozwik, K., Schrimpf, M., Kanwisher, N., Dicarlo, James. 2019. To find better neural network models of human vision, find better neural network models of primate vision. BioRxiv.

[7] Conwell, Colin, et al. "What can 5.17 billion regression fits tell us about artificial models of the human visual system?." SVRHM 2021 Workshop@ NeurIPS. 2021.

[8] Heeger, D., Ress, D. 2002. What does fMRI tell us about neuronal activity? Nature reviews neuroscience.

---

> ### Comment · Reviewer_nyjr · 2023-08-14
> **clarification regarding BrainScore layer selection**
>
> From the authors response above:
> > Our neural data fitting procedure followed the original Brain-Score [1] procedure precisely, “The final neural predictivity score for the target brain region is computed as the mean across all train-test splits.” In other words, we (and the standard Brain-Score) did not use a held-out validation set to select the most predictive layer for each model.
>
> I believe that this is incorrect, at least based on the current Brain-Score website. The layer selection on BrainScore uses the "public" datasets for each region. This is the "Layer Commitment" in the "BaseModel"->"BrainModel" conversion for each dataset. I believe there is also an option to pre-select specific layers if an anatomical mapping is built into the architecture. Quote from the BrainScore website:
> > BaseModel layers have to be committed to cortical regions. For BaseModels that are automatically translated into BrainModels this is done on separate public data. The same layers are thus used when recording from the same cortical region, e.g. always the same layer for V1 instead of different layers per benchmark.
>
> Nevertheless, the new analysis by the authors selecting the layers based on validation data and testing on test test seems like the correct measure that should be reported in the final paper (and if these results are the same trends, I'm not sure if it is worth including the results from the previous analysis which had "double dipping" for the layer selection).

---

> > ### Author Response · Authors · 2023-08-14
> > **Response**
> >
> > Thanks so much for the response and the original review, which we believe greatly strengthened our paper.
> >
> > The point of confusion here is whether we adopt the method from the Brain-Score paper vs. the Brain-Score website, which (as is pointed out) has private test sets. In our original submission we followed the method from the Brain-Score paper, and with our rebuttal we added results that follow the method from the website.
> >
> > We can include both in the final version of the paper (moving results from the Brain-Score paper method to the Appendix) and make this distinction clear. Is that OK?

---

> > > ### Comment · Reviewer_nyjr · 2023-08-14
> > >
> > > Got it. That makes sense. Yes I think the proposal of moving the results from the Brain-Score paper to the supplement would be fine. Thanks again for re-running the analysis with the improved layer-selection procedure.

---

> > > > ### Author Response · Authors · 2023-08-14
> > > > **Response**
> > > >
> > > > Thank you for your time!

---

### Decision · Program_Chairs · 2023-09-21

**Decision:**

Accept (poster)

**Comment:**

The paper investigates the finding that more recent strong image recognition models have become worse models of monkey IT cortex and proposes a "neural harmonizer" to align model and brain responses. The reviewers raised some concerns about the exact contributions and potential controls with additional datasets, which the authors addressed during the discussion period. As a result of this interaction, the reviewers agree that the paper makes a valuable contribution and should be accepted.

As a note to the authors: To contextualize your results, you are expected to include the results on the BrainScore dataset which came out of the discussion with reviewer nyjr.